# Implanting oxophilic metal in PtRu nanowires for hydrogen oxidation catalysis

Zhongliang Huang[1,11], Shengnan Hu[1,11], Mingzi Sun[2], Yong Xu[3] ✉, Shangheng Liu[1], Renjie Ren[4], Lin Zhuang[4], Ting-Shan Chan[5], Zhiwei Hu[6], Tianyi Ding[1], Jing Zhou[7], Liangbin Liu[1], Mingmin Wang[1], Yu-Cheng Huang[8], Na Tian[1], Lingzheng Bu[9] ✉, Bolong Huang[2] ✉ & Xiaoqing Huang[1,10] ✉

Bimetallic PtRu are promising electrocatalysts for hydrogen oxidation reaction in anion exchange membrane fuel cell, where the activity and stability are still unsatisfying. Here, PtRu nanowires were implanted with a series of oxophilic metal atoms (named as i-M-PR), significantly enhancing alkaline hydrogen oxidation reaction (HOR) activity and stability. With the dual doping of In and Zn atoms, the i-ZnIn-PR/C shows mass activity of 10.2 A $mg_{Pt+Ru}^{-1}$ at 50 mV, largely surpassing that of commercial Pt/C (0.27 A $mg_{Pt}^{-1}$) and PtRu/C (1.24 A $mg_{Pt+Ru}^{-1}$). More importantly, the peak power density and specific power density are as high as 1.84 W $cm^{-2}$ and 18.4 W $mg_{Pt+Ru}^{-1}$ with a low loading (0.1 mg $cm^{-2}$) anion exchange membrane fuel cell. Advanced experimental characterizations and theoretical calculations collectively suggest that dual doping with In and Zn atoms optimizes the binding strengths of intermediates and promotes CO oxidation, enhancing the HOR performances. This work deepens the understanding of developing novel alloy catalysts, which will attract immediate interest in materials, chemistry, energy and beyond.

Anion exchange membrane fuel cells (AEMFCs), which can power vehicles with zero $CO_2$ emissions, have attracted increasing attention in the pursuit of a sustainable energy future society[1,2]. As the anode reaction, the hydrogen oxidation reaction (HOR) performance is insufficient for industrial requirements, which limits the practical application of AEMFCs. Over the past decades, substantial efforts have been devoted to the fabrication and modification of efficient electrocatalysts for improving HOR performance[3–6]. Nevertheless, the high-efficiency process of HOR is impeded by the current disadvantages

including unsatisfying stability and activity of electrocatalysts[7,8]. In particular, the HOR kinetics in alkaline conditions is generally 2–3 orders of magnitude lower than in acidic conditions[2,7–11]. Consequently, the design of efficient catalysts for alkaline HOR has emerged as the new frontier in hydrogen electrocatalysis yet remains a great challenge.

Platinum (Pt) has been widely used as HOR catalysts due to its strong affinity to $H_2$[9,10]. Nevertheless, ideal HOR catalysts should be capable of appropriate *H adsorption and strengthened *OH

[1]State Key Laboratory for Physical Chemistry of Solid Surfaces, College of Chemistry and Chemical Engineering, Xiamen University, Xiamen 361005, China. [2]Department of Applied Biology and Chemical Technology, The Hong Kong Polytechnic University, Hung Hom, Kowloon, Hong Kong SAR 999077, China. [3]Nano-X Vacuum Interconnected Nano-X Vacuum Interconnected Workstation, Suzhou Institute of Nano-Tech and Nano-Bionics (SINANO), Chinese Academy of Sciences (CAS), Suzhou, Jiangsu 215123, China. [4]College of Chemistry and Molecular Sciences, Hubei Key Lab of Electrochemical Power Sources, Wuhan University, Wuhan 430072, China. [5]National Synchrotron Radiation Research Center, 101 Hsin-Ann Road, Hsinchu 30076, Taiwan. [6]Max Planck Institute for Chemical Physics of Solids, Nothnitzer Strasse 40, Dresden 01187, Germany. [7]Shanghai Institute of Applied Physics, Chinese Academy of Sciences, Shanghai 201800, China. [8]Department of Electrophysics, National Yang Ming Chiao Tung University, Hsinchu 30010, Taiwan. [9]College of Energy, Xiamen University, Xiamen 361102, China. [10]Innovation Laboratory for Sciences and Technologies of Energy Materials of Fujian Province (IKKEM), Xiamen 361005, China. [11]These authors contributed equally: Zhongliang Huang, Shengnan Hu. ✉e-mail: yongxu@gdut.edu.cn; lzbu@xmu.edu.cn; bhuang@polyu.edu.hk; hxq006@xmu.edu.cn

adsorption. To this end, great efforts have been devoted to the modification of Pt with other metals to regulate its binding strength towards *H and *OH. In particular, Ru has attracted great interest due to its similar affinity to $H_2$ but lower price than Pt, which has been used to composite with Pt for HOR[11,12]. However, the precise modulation of the electronic properties of Pt and Ru for balancing the adsorption abilities toward *H and *OH remains a great challenge. Besides, $H_2$ produced from the reforming process inevitably contains a trace amount of CO, which can strongly adsorb on the surface of Pt and Ru, leading to the poisoning of active sites for HOR[13–15]. A feasible strategy is to introduce heteroatoms to weaken the binding strength of CO and/or promote *CO conversion to $CO_2$. In other words, an ideal PtRu-based catalyst for practical HOR should well balance the adsorption abilities to *H, *OH, and *CO[16,17], which is formidably challenging yet.

In this work, we have proposed a universal strategy for introducing oxophilic metal atoms (e.g., Mn, Fe, Co, Ni, Cu, Zn, Ga, and In) into PtRu nanowires (noted as i-M-PR). Microstructure analysis, X-ray absorption fine structure spectroscopy, cyclic voltammogram (CV), isotopic measurement, and theoretical calculations collectively show that the introduction of In atoms can promote HOR activity, while Zn atoms can significantly enhance the stability and resistance to CO-poisoning. Significantly, the activity, stability, and resistance to CO-poisoning of PR can be simultaneously enhanced when combined dual doping with In and Zn atoms. In particular, the implantation of oxophilic metal atoms can lower the $d$-band center of surface Pt site and thus reduce the binding strengths of *H and *CO, while the $e_g$-$t_{2g}$ splitting of $4d$ orbitals for Ru sites reduces electron density near Fermi level and facilitates *OH adsorption, as a result of enhanced HOR activity and resistance to CO-poisoning. Consequently, the mass activity of optimal i-ZnIn-PR/C reaches 10.2 A $mg_{Pt+Ru}^{-1}$ at 50 mV, which is much higher than that of commercial Pt/C (0.27 A $mg_{Pt}^{-1}$) and PtRu/C (1.24 A $mg_{Pt+Ru}^{-1}$). Moreover, i-ZnIn-PR/C exhibits improved stability at 100 mV (vs. RHE) with a slight current decay of 5.3% after 10,000 s. An AEMFCs with low noble metal loading of i-ZnIn-PR/C as anode catalyst and commercial Pt/C as cathode catalyst achieved a peak power density (PPD) and specific power density (SPD) of 1.84 W $cm^{-2}$ and 18.4 W $mg_{Pt+Ru}^{-1}$. Impressively, i-ZnIn-PR/C displays superior resistance to CO-poisoning to commercial PtRu/C and Pt/C, with a reserved current of 84.9% compared with the initial value after 5000 s.

## Results

Pristine PR was synthesized via a chemical approach, while the introduction of foreign metals was simply realized by adding corresponding precursors (Fig. 1a). High-angle annular dark-field scanning transmission electron microscopy (HAADF-STEM) image showed that PR presented as one-dimensional (1-D) nanowire with a mean diameter of 1.2 ± 0.3 nm (Supplementary Figs. 1a, b). The lattice distance of 0.224 nm in high-resolution TEM (HRTEM) image was close to that of Pt (111) facet (Supplementary Fig. 1c). Energy dispersive X-ray spectroscopy (EDS) mapping indicated that the molar ratio of Pt and Ru was 86.9/13.1 (Supplementary Fig. 1d). After introducing In into PR, the 1-D morphology was reserved even when the molar percentage of In was increased from 2% (i-In$_2$-PR) to 6.3% (i-In$_{6.3}$-PR) (Fig. 1b and Supplementary Fig. 2 and Supplementary Table 1). The peaks in the X-ray diffraction (XRD) pattern could be indexed as the face-centered cubic ($fcc$) phase of Pt (JCPDS No. 04−0802) (Fig. 1c). Compared to pure Pt, the peaks in the XRD pattern of PR shifted to high angle, which was attributed to the lattice contraction after Ru introduction. Moreover, a negative shift of Pt (111) peak was observed after introducing In (3.9%) into PR (noted as i-In-PR), which was ascribed to the larger radius of In than that of Ru. Correspondingly, the lattice distance in aberration-corrected HAADF-STEM images increased from 0.224 to 0.238 nm after introducing In into PR (Fig. 1d), being consistent with the negative peak shift in the XRD pattern. Besides, significant lattice distortion was observed in i-In-PR, suggesting that the introduction of In could induce

distortion formation. Furthermore, EDS element mapping implied that In had been successfully introduced into PR (Supplementary Fig. 3). In addition, X-ray absorption near-edge spectroscopy (XANES) and extended X-ray absorption fine structure (EXAFS) measurements were employed to study the electronic and coordination structure of i-In-PR. As shown in Fig. 1e, the Pt $L_3$-edge XANES spectrum of PR and i-In-PR displayed similar features to that of Pt foil, indicating that Pt mainly presented the metallic state in PR and i-In-PR, which was further validated by the presence of Pt−Pt (Ru/In) coordination in EXAFS spectrum (Fig. 1f) and wavelet transform pattern (Supplementary Fig. 4a)[18]. Analysis of Ru $K$-edge XANES spectrum suggested that Ru in i-In-PR presented as oxidative state (Ru$^{\delta+}$, $0 < \delta < 4$) (Fig. 1g)[19], which was further confirmed by the presence of Ru−O in EXAFS spectrum (Fig. 1h and Supplementary Fig. 4b). Note that the radius of Ru−O in i-In-PR was larger than that of $RuO_2$ and PR, which was attributed to the formation of Ru−O−M (M = In or Zn) with stronger electronegativity than Ru[20]. Moreover, the oxidative state of In in i-In-PR was further validated by X-ray photoelectron spectroscopy (XPS) (Supplementary Fig. 5)[21]. Impressively, other metals including Mn, Fe, Co, Ni, Cu, Zn, and Ga could also be introduced into PR with the present strategy (Supplementary Figs. 6−13). The normalized Pt $L_3$-edge XANES and EXAFS spectra of i-M-PR (M = Mn, Fe, Co, Ni, Cu, Zn, Ga) exhibited similar features to that of Pt foil, suggesting that Pt in i-M-PR mainly presented the metallic state (Fig. 1i, j and Supplementary Fig. 14). The reservation of 1-D i-M-PR nanowires implied the universality of this strategy for introducing foreign atoms.

To evaluate the influence of different In molar percentages on catalytic performance, i-In$_x$-PR ($x$ represents molar percentage) were loaded on Vulcan XC-72R carbon and used as catalysts for HOR (Supplementary Fig. 15 and Supplementary Tables 1, 2). Screening experiments displayed that the optimal molar percentage of In was 3.9% (i.e., i-In-PR/C) (Supplementary Fig. 16), and therefore the theoretical molar percentages of other metals (i.e., Mn, Fe, Co, Ni, Cu, Zn and Ga) were kept at ~3.9% (Supplementary Fig. 17 and Supplementary Tables 1, 2). The CV curves of various catalysts were compared in 0.1 M $HClO_4$ (Supplementary Fig. 18), while the HOR activity of the catalysts was evaluated in $H_2$-saturated KOH (0.1 M). Analysis of HOR polarization curves suggested that i-In-PR/C exhibited superior HOR activity to other catalysts (Fig. 2a), which was validated by its larger kinetic current density ($J_k$) based on the Koutecky-Levich equation (Fig. 2b)[22]. We compared the CO stripping, underpotential deposition H ($H_{upd}$), and Cu ($Cu_{upd}$) experiments, and the results show that the CO stripping experiment used in this study is reasonable for calculating ECSA (Supplementary Fig. 19). To quantitatively compare the activities of different catalysts, the mass activity and specific activity were normalized with respect to both the electrochemically active surface areas (ECSA, determined by CO-stripping experiments) and the loading amounts of noble metals (Fig. 2c, Supplementary Fig. 20 and Supplementary Table 3). As shown in Fig. 2d, the mass activity and specific activity of i-In-PR/C reached 8.51 A $mg_{Pt+Ru}^{-1}$ and 12.21 mA $cm^{-2}$, respectively, much higher than those of i-Mn-PR/C, i-Fe-PR/C, i-Co-PR/C, i-Ni-PR/C, i-Cu-PR/C, i-Zn-PR/C, and i-Ga-PR/C (Supplementary Table 4). Besides, 99.3% of the initial current was reserved for i-Zn-PR/C at 4000 s at 100 mV in the durability test, which was much higher than those of other references (Fig. 2e and Supplementary Fig. 21). Additionally, the resistance to CO-poisoning was evaluated by bubbling 1000 ppm $CO/H_2$ into 0.1 M KOH solution, where the resistance to CO-poisoning followed an order of i-Zn-PR/C > i-Mn-PR/C > i-Cu-PR/C > i-Co-PR/C > i-Fe-PR/C > i-Ni-PR/C > i-Ga-PR/C > i-In-PR/C (Fig. 2f and Supplementary Fig. 22). The above results imply that the introduction of In atom is able to promote the HOR activity, while Zn atoms can significantly enhance the stability and resistance to CO-poisoning (Fig. 2g).

Given the significance of In and Zn on HOR performance, we simultaneously introduced different molar ratios of In and Zn into PR

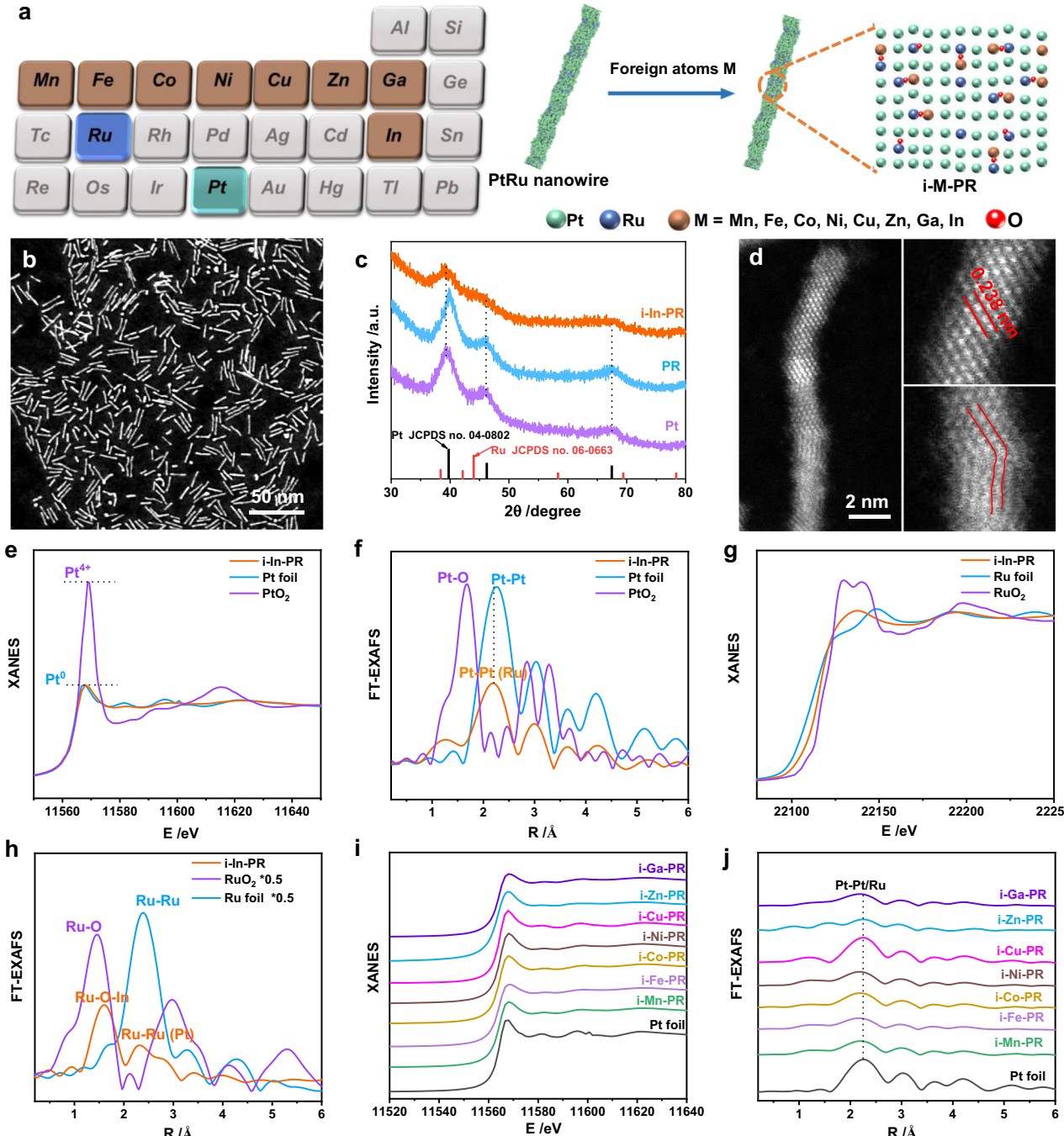

**Fig. 1 | Characterizations of i-M-PR. a** Scheme for introducing foreign atoms into PR. **b** HAADF-STEM image, (**c**) XRD pattern, and **d** Aberration-corrected HAADF-STEM images. **e** Pt $L_3$-edge XANES spectra, (**f**) Pt $L_3$-edge FT-EXAFS spectra, (**g**) Ru $K$-edge XANES spectra, and (**h**) Ru $K$-edge FT-EXAFS spectra of i-In-PR. **i** Pt XANES and (**j**) FT-EXAFS spectra of i-M-PR at $L_3$-edge.

(Supplementary Fig. 23, 24) and Supplementary Table 5 for alkaline HOR. Commercial Pt/C (40 wt.%) and PtRu/C (60 wt.%) were used as references (Supplementary Fig. 25). It was noted that the mass activity followed an order of i-Zn$_1$In$_{1.1}$-PR/C > i-Zn$_{1.8}$In$_1$-PR/C > i-Zn$_1$In$_{2.2}$-PR/C (Supplementary Fig. 26), and i-Zn$_1$In$_{1.1}$-PR/C was therefore selected as the optimal catalyst (noted as i-ZnIn-PR/C). Based on HOR polarization curves, i-ZnIn-PR/C displayed superior HOR activity to other references (Fig. 3a). Then, we investigated the HOR polarization curves on Pt/C, PtRu/C, and i-ZnIn-PR/C as a function of the rotation rate. It was noted that the current density increased with increasing rotation rate due to the facilitation of mass transport (Supplementary Fig. 27), and

the similar slopes for i-ZnIn-PR/C (12.6 cm$^{-2}$ mA$^{-1}$ rpm$^{1/2}$) to the theoretical value further proved the H$_2$ mass transport control process (Supplementary Fig. 27d)[23]. No anodic current was detected in N$_2$-saturated electrolyte, indicating that the current was originated from HOR (Fig. 3a). The exchange currents ($I_0$) and area specific exchange current density ($J_{0,s}$) by fitting with Butler–Volmer (BV) equation and micro-polarization region (MR) implied the superior HOR activity of i-ZnIn-PR/C to other references (Supplementary Figs. 28, 29 and Supplementary Table 4). The fitted curve of the Tafel plot and the charge transfer coefficient ($\alpha$) extrapolated based on the Butler-Volmer fitting are shown in Supplementary Fig. 30 and Supplementary Table 4.

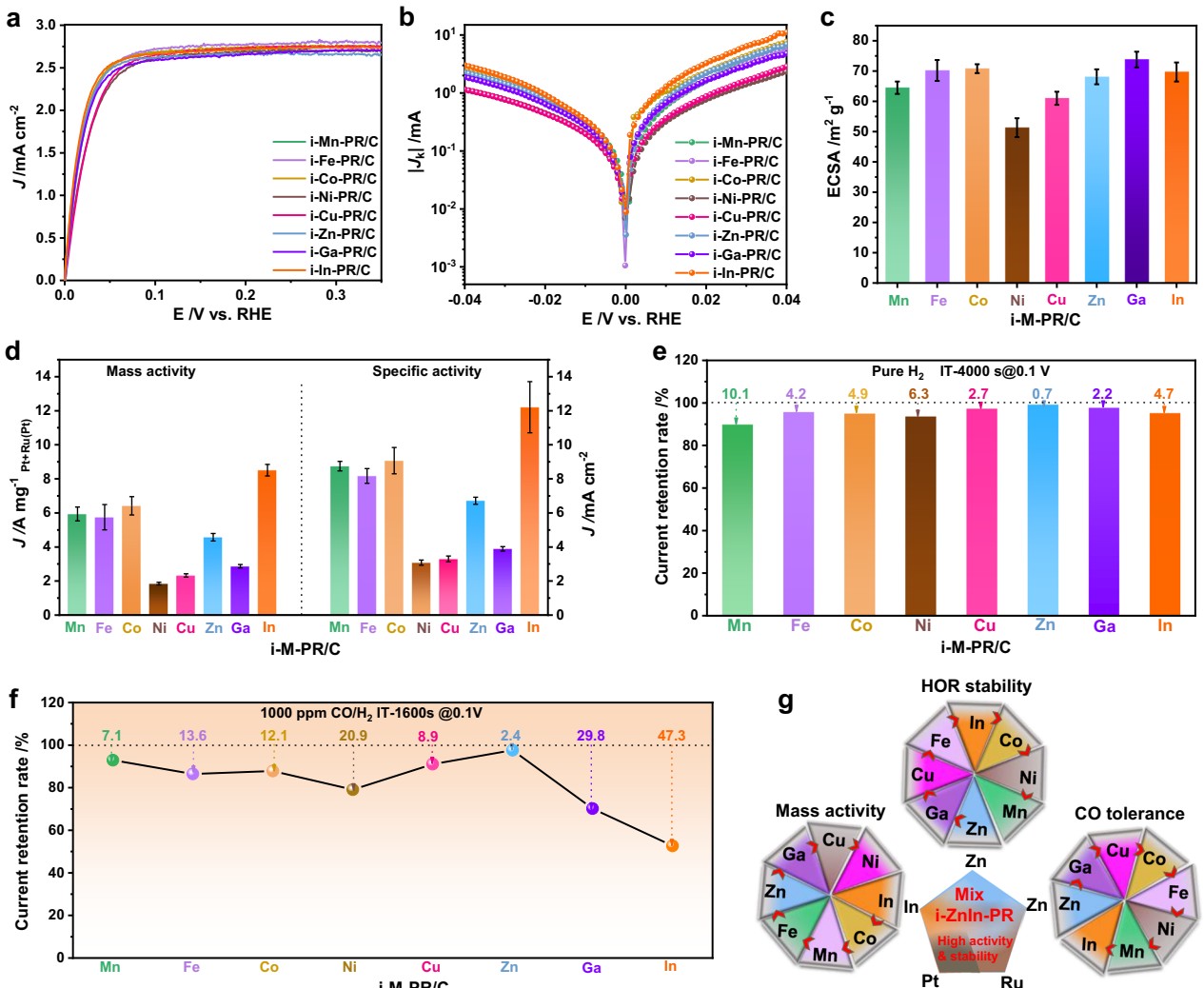

**Fig. 2 | HOR performance. a** HOR polarization curves recorded in $H_2$-saturated 0.1 M KOH with a sweeping rate of 50 mV s$^{-1}$ at a rotation rate of 1600 rpm. **b** Representative HOR Tafel plots of kinetic current density. **c** ECSAs of i-M-PR/C at 50 mV (vs. RHE). **d** ECSA normalized specific activity and mass activity at 50 mV (vs.

RHE). **e** Current retention rate of various catalysts in $H_2$-saturated 0.1 M KOH solution at 100 mV (vs. RHE). **f** CO tolerance of i-M-PR/C evaluated at 100 mV (vs. RHE). **g** Scheme of the significance of oxophilic metal atoms on HOR performance. The error bars in (**c**, **d**) were obtained by three parallel experiments.

Impressively, the mass activity of i-ZnIn-PR/C reached 10.2 A mg$_{Pt+Ru}$$^{-1}$, which was 37.8 and 8.2 times higher than those of commercial Pt/C and PtRu/C, respectively (Fig. 3b). Moreover, the specific activity normalized by ECSA of i-ZnIn-PR/C was 27.3 and 7.3 times higher than that of commercial Pt/C and PtRu/C (Supplementary Table 4). Note that the HOR performance of i-ZnIn-PR/C has outperformed many reported catalysts (Fig. 3c and Supplementary Table 6). Furthermore, when i-ZnIn-PR/C was used as HOR catalyst, the current decreased by 5.3% after 10,000 s, which was lower than those of PR/C (6.1%) and commercial PtRu/C (65.8%) (Fig. 3d). Note that the current decreased by 45.9% for commercial Pt/C after only 5000 s. The enhanced stability of i-ZnIn-PR/C for alkaline HOR was further evaluated by collecting the polarization curves after 2000 cycles in the accelerated durability test (ADT) and chronoamperometry, where the curve after 2000 cycles almost overlapped with the initial one (Supplementary Fig. 31). Besides, the morphology of nanowires was largely reserved in the used i-ZnIn-PR/C after ADT, while severe aggregation was observed for commercial Pt/C and PtRu/C (Supplementary Figs. 32, 33), suggesting that i-ZnIn-PR/C could serve as a highly active and stable catalyst for alkaline HOR. Additionally, the limiting current density of i-ZnIn-PR/C at 100 mV (vs. RHE) slightly decreased by ~1.8% after introducing 1000 ppm CO, which was much smaller than those of PR/C (9.2%),

commercial PtRu/C (5.4%) and commercial Pt/C (30.0%) (Supplementary Fig. 34). Besides, the currents decreased by 15.1% for i-ZnIn-PR/C after 5000 s, while the currents strongly decreased by 43.6% and 52.4% for PR/C and commercial PtRu/C, respectively, after only 4000 s, indicating the superior resistance to CO-poisoning of i-ZnIn-PR/C to other references (Fig. 3e). Note that the resistance to CO-poisoning of i-ZnIn-PR/C has surpassed many reported catalysts (Supplementary Fig. 35 and Supplementary Table 7). Impressively, i-ZnIn-PR/C exhibits excellent AEMFCs activity with a low loading amount of noble metal. Fig. 3f, h shows the comparison of AEMFCs polarization and power density curves at $H_2/O_2$ and $H_2/Air$ ($CO_2$-free) for different anode catalysts. For $H_2/O_2$, the PPD of AEMFCs with i-ZnIn-PR/C reaches 1.84 W cm$^{-2}$ at 4.6 A cm$^{-2}$, which is higher than the AEMFCs with commercial Pt/C (PPD of 0.85 W cm$^{-2}$ at 2.0 A cm$^{-2}$) and PtRu/C (PPD of 1.11 W cm$^{-2}$ at 2.2 A cm$^{-2}$). Note that the anode noble metal utilization of this AEMFCs is 18.4 W mg$_{Pt+Ru}$$^{-1}$, one of the highest values among the reported membrane-electrode assemblies (Fig. 3g)[24]. Additionally, similar results was obtained in $H_2/Air$ ($CO_2$-free) cell (Fig. 3h). The i-ZnIn-PR/C AEMFCs delivers PPD of 1.19 W cm$^{-2}$ at 3.0 A cm$^{-2}$ under $H_2$-Air ($CO_2$-free), which is higher than the AEMFCs with commercial Pt/C (PPD of 0.67 W cm$^{-2}$ at 1.6 A cm$^{-2}$) and PtRu/C (PPD of 0.87 W cm$^{-2}$ at 2.0 A cm$^{-2}$).

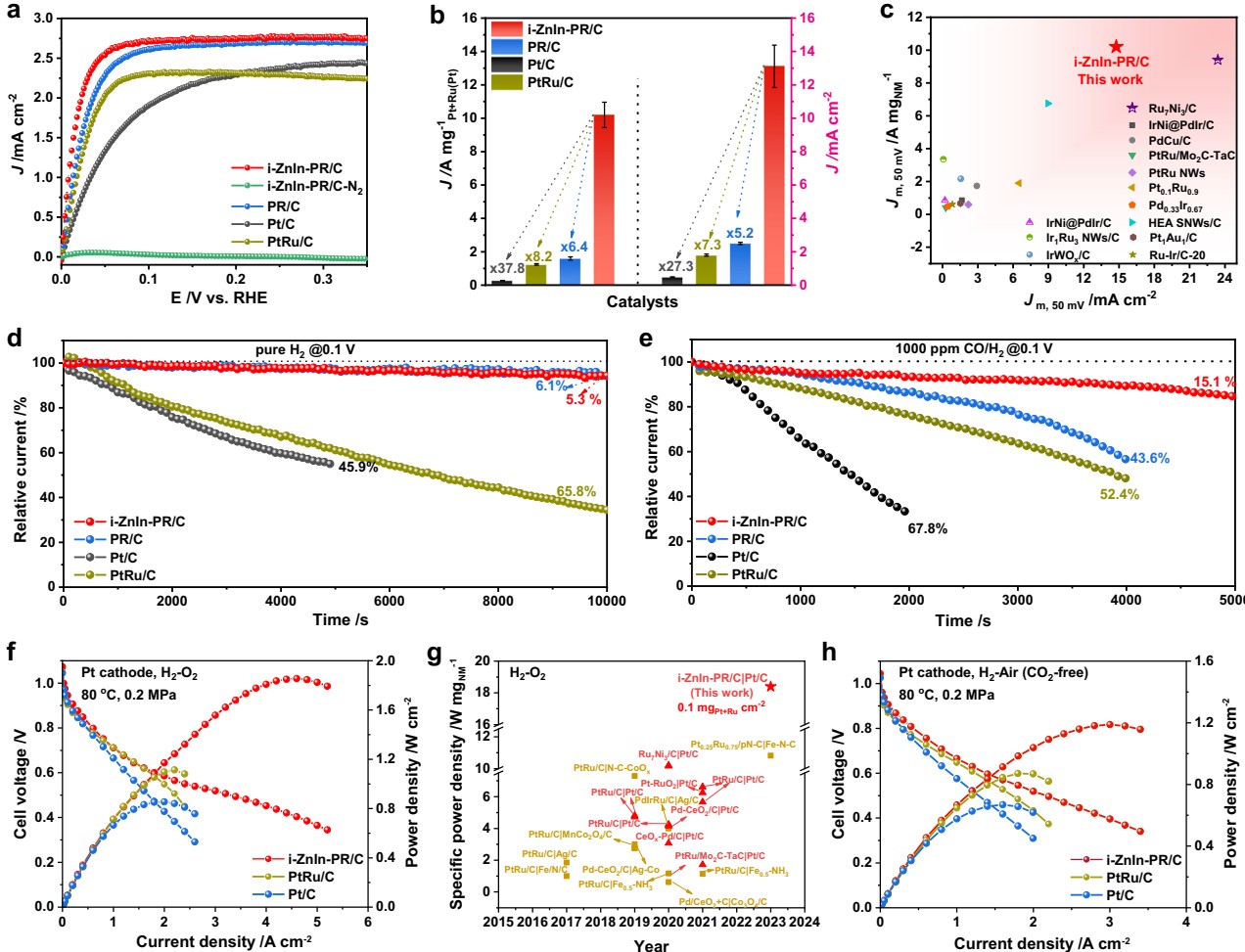

**Fig. 3 | HOR performance evaluation of Pt-based catalysts. a** HOR polarization curves were recorded in $H_2$-saturated 0.1 M KOH with a sweeping rate of 50 mV s$^{-1}$ at a rotation rate of 1600 rpm. **b** ECSA normalized specific activity and mass activity at 50 mV (vs. RHE). **c** Comparison of mass activity at 50 mV (vs. RHE) and specific activity between i-ZnIn-PR/C and other reported catalysts. **d** Durability test for i-ZnIn-PR/C and references in an $H_2$-saturated 0.1 M KOH solution at 100 mV (vs. RHE). **e** CO tolerance test for different catalysts at 100 mV (vs. RHE). **f** Fuel cell performance test of $H_2/O_2$ AEMFCs with commercial Pt/C (0.1 mg$_{Pt}$ cm$^{-2}$), PtRu/C (0.1 mg$_{Pt+Ru}$ cm$^{-2}$) and i-ZnIn-PR/C (0.1 mg$_{Pt+Ru}$ cm$^{-2}$) in anode and Pt/C (0.4 mg$_{Pt}$ cm$^{-2}$) in cathode. **g** Comparison of the anode noble metal utilization for $H_2-O_2$ AEMFCs. **h** Fuel cell performance test of $H_2$/Air (CO$_2$-free) AEMFCs with commercial Pt/C (0.1 mg$_{Pt}$ cm$^{-2}$), PtRu/C (0.1 mg$_{Pt+Ru}$ cm$^{-2}$) and i-ZnIn-PR/C (0.1 mg$_{Pt+Ru}$ cm$^{-2}$) in anode and Pt/C (0.4 mg$_{Pt}$ cm$^{-2}$) in cathode. The error bars in (**b**) was obtained by three parallel experiments.

Given the significance of In and Zn on HOR performance, more characterizations were conducted to reveal the structures of i-ZnIn-PR/C. HADDF-STEM image displayed that both the morphology and dimension were largely reserved after introducing In and Zn (Fig. 4a). Compared to i-In-PR, the positive peak shift in the XRD pattern of i-ZnIn-PR was ascribed to Zn introduction (Fig. 4b). Moreover, the successful introduction of In and Zn was validated by EDS mapping (Supplementary Fig. 36). The lattice distance of 0.224 nm in HRTEM image was close to that of pristine PR (Fig. 4c). Similarly, obvious lattice distortion and defects were observed in HRTEM images (Fig. 4d, e). Pt mainly presented as the metallic state (Pt$^0$), while Ru, In, and Zn mainly presented as an oxidative state based on XPS (Supplementary Fig. 37) and XANES spectra of i-ZnIn-PR at Pt $L_3$-edge, Ru $K$-edge, In $K$-edge and Zn $K$-edge (Fig. 4f–m). The presence of Pt−Pt (M) (M = Ru, In or Zn) coordination in Pt $L_3$-edge EXAFS spectrum (Fig. 4g, Supplementary Fig. 38, and Supplementary Table 8) and Ru−O coordination in Ru $K$-edge EXAFS spectrum (Fig. 4i, Supplementary Fig. 39, and Supplementary Table 9) further confirmed the metallic state of Pt and the oxidative state of Ru in i-ZnIn-PR. Moreover, analysis of the XANES and EXAFS spectra of i-ZnIn-PR at Zn $K$-edge and In $K$-edge suggests that In and Zn mainly present as the oxidative state (Fig. 4j–m). Specifically, in

addition to the presence of Zn−O coordination and In−O coordination, the appearance of Zn−Pt coordination (Fig. 4k, Supplementary Fig. 40 and Supplementary Table 10) and In−Pt coordination (Fig. 4m and Supplementary Fig. 41 and Supplementary Table 11) indicate that the Zn and In atoms in i-ZnIn-PR are coordinated with O in the form of Zn−O−Ru/In−O−Ru and Pt in the form of Zn−Pt/In−Pt[25–28].

To reveal the mechanism for enhanced HOR performance after introducing Zn and In atoms, we initially examined the adsorption ability towards hydrogen by CV curves. It was noted that introducing In and Zn atoms could strongly weaken the adsorption ability towards $H_2$ (Fig. 5a). Moreover, we plotted the diagram between the area of weak hydrogen adsorption and the logarithm of the exchange current density to demonstrate the correlation between hydrogen adsorption and HOR activity (Fig. 5b and Supplementary Fig. 42). The positively linear correlation indicated that introducing In and Zn atoms could weaken the hydrogen binding energy (HBE) and promote HOR activity. Moreover, the potential influence of In and Zn atoms on hydroxide binding energy (OHBE) was studied by isotopic experiment using deuterated KOH (e.g., KOD). Compared to KOH, the polarization curve of i-ZnIn-PR/C collected in KOD shifts to low potential (Fig. 5c), which was ascribed to its smaller ionic product constant[29]. The potentials for

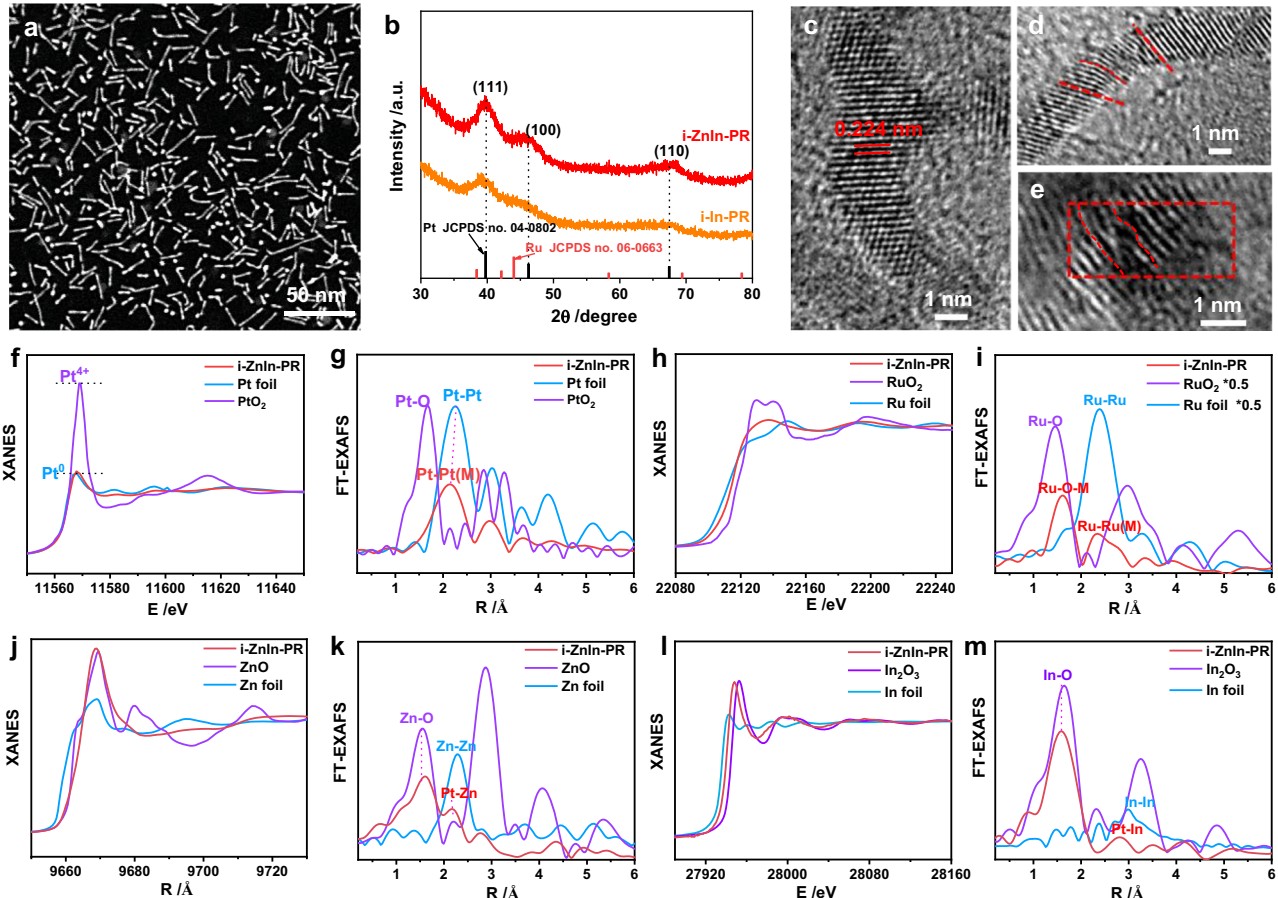

**Fig. 4 | Structural characterizations of i-ZnIn-PR. a** HAADF-STEM image, (**b**) XRD pattern, (**c**–**e**) HRTEM images, (**f**) Pt $L_3$-edge XANES spectra, (**g**) Pt $L_3$-edge FT-EXAFS spectra, (**h**) Ru $K$-edge XANES spectra, (**i**) Ru $K$-edge FT-EXAFS spectra, (**j**) Zn $K$-edge XANES spectra, (**k**) Zn $K$-edge FT-EXAFS spectra, (**l**) In $K$-edge XANES spectra, (**m**) In $K$-edge FT-EXAFS spectra.

reaching 2.5 mA cm$^{-2}$ in 0.1 M KOD and 0.1 M KOH were 0.064 V and 0.042 V, respectively, implying the superior oxidation resistance of i-ZnIn-PR/C in KOD to KOH. Furthermore, analysis of the Tafel diagram revealed that the HOR kinetics of i-ZnIn-PR/C and PR/C in KOD were much slower than those in KOH (Fig. 5d and Supplementary Fig. 43), suggesting that HOR performance was strongly dependent on OHBE. Given the superior HOR performance of i-ZnIn-PR/C in 0.1 M KOD to that of PR/C in 0.1 M KOH, we concluded that the introduction of Zn and In atoms could significantly enhance OHBE and improve the HOR activity (Fig. 5e). Compared with PR/C, the Pt 4$f$ XPS spectrum of i-ZnIn-PR/C moves towards lower energy, while the peak of Ru 3$p$ XPS spectrum moves towards higher energy (Supplementary Fig. 44), indicating that electrons may be transferred from Ru to Pt in i-ZnIn-PR/C. Compared with PR/C (−3.71 eV), the $d$-band center of i-ZnIn-PR/C moves downward (Supplementary Fig. 45), indicating that the introduction of Zn and In was conducive to weakening the adsorption of H and promoting the adsorption of OH intermediates. Furthermore, in situ diffuse reflectance infrared Fourier transform spectroscopy (DRIFTS) and in situ electrochemical Raman spectroscopy were performed to reveal the mechanism for enhancing the resistance of i-ZnIn-PR/C to CO poisoning (Supplementary Fig. 46). When commercial Pt/C was exposed to KOH solution saturated with 100 ppm CO/H$_2$ (Fig. 5f), the bands at 1384, 1576 and 2910 cm$^{-1}$ in the DRIFTS spectra of commercial Pt/C were ascribed to *COOH, while the band at 1900 cm$^{-1}$ could be indexed as bridged-adsorbed CO (B-*CO)[30]. In contrast, no obvious peaks of B-*CO and *COOH were observed in the DRIFTS spectra of i-ZnIn-PR/C after exposure to CO, while the bands at 1538, 1393 and 1306 cm$^{-1}$ corresponded to CO$_3^{2-}$ (Fig. 5g), indicating that *COOH might be converted

into CO$_3^{2-}$ with *OH on the catalyst surface[30]. In the Raman spectra of i-ZnIn-PR/C, a peak corresponding to *OH appeared at 985 cm$^{-1}$ in the presence of H$_2$ at 0.1–0.6 V (left of Fig. 5h)[31], which was further validated by the peak shift from 983 cm$^{-1}$ to 744 cm$^{-1}$ when H was replaced by D in isotope experiment (Supplementary Fig. 47)[32]. After introducing CO into the solution, the disappearance of *OH peaks confirmed that the adsorbed *CO could react with *OH (right of Fig. 5h). Based on the results from in situ DRIFTS and Raman spectroscopy, we concluded that the absorbed *CO on i-ZnIn-PR/C would combine with *OH to form CO$_2$ through the intermediate of *COOH, as a result of improved resistance to CO-poisoning (Fig. 5i).

Density functional theory (DFT) calculations were conducted to investigate the enhanced HOR performance over i-ZnIn-PR with oxophilic metal atoms. Compared to PR, evident lattice distortions were observed after introducing In and Zn atoms, which was consistent with the experimental results of lattice space expansion and lattice stripe distortion (Fig. 6a, b). We further compared the electronic structures of i-ZnIn-PR and PR through the projected partial density of states (PDOSs). As shown in Fig. 6c, Pt-5$d$ and Ru-4$d$ orbitals contributed to the electron density near the Fermi level ($E_F$) as the main active sites for i-ZnIn-PR. Meanwhile, the broad In-5$p$ orbitals crossed the $E_F$, which further improved the electron depletion efficiency from i-ZnIn-PR to the intermediates. Zn-3$d$ orbitals exhibited a sharp peak at $E_V$ − 6.67 eV ($E_V$ = 0 eV), while O-2$p$ orbitals and In-5$p$ orbitals pinned the electronic structures of both Pt and Ru, supplying highly stable valence states during the HOR. In contrast with i-ZnIn-PR, the overall $d$-band center of PR downshifted with the decreased electroactivity (Fig. 6d). Moreover, the $e_g$-$t_{2g}$

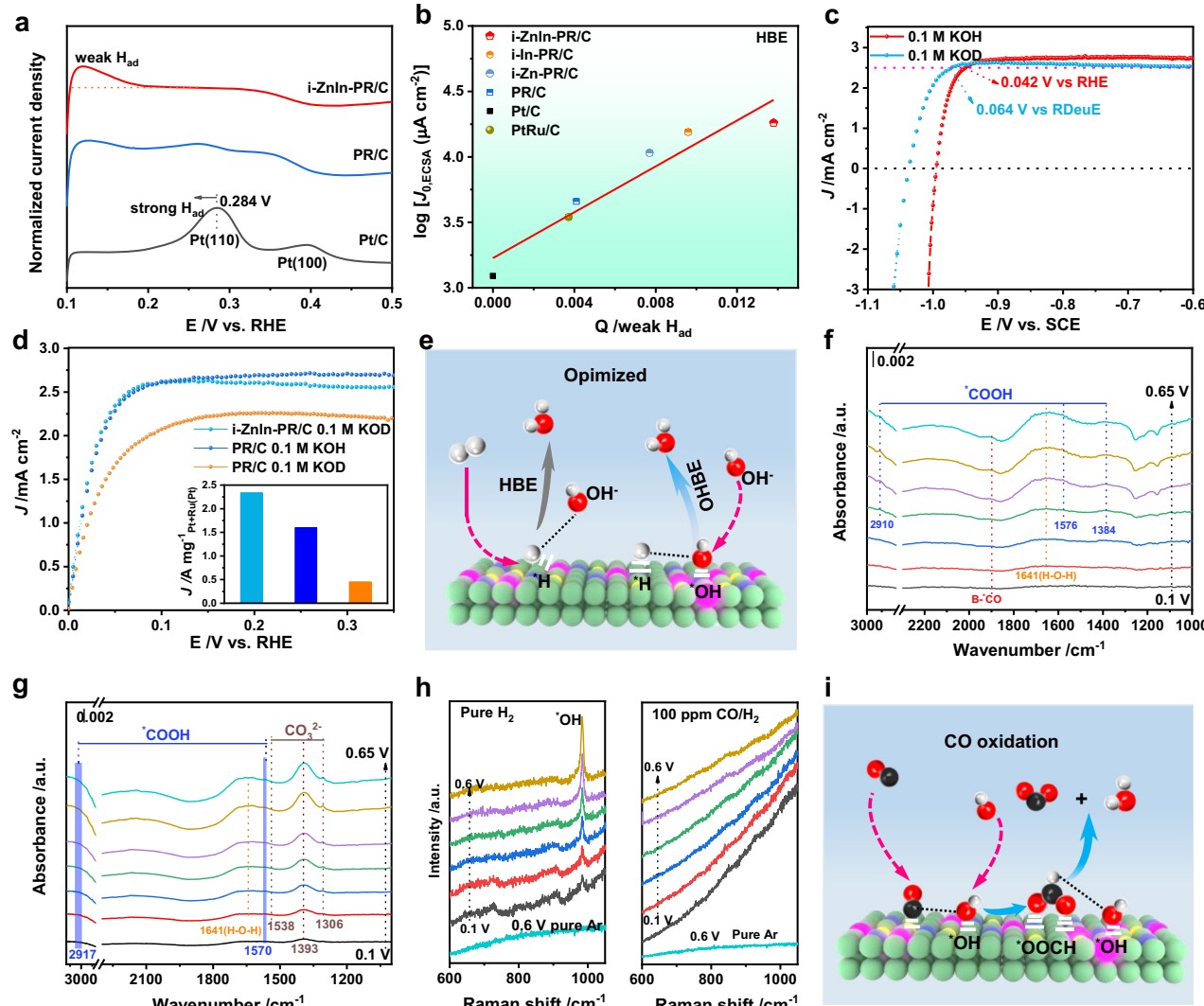

**Fig. 5 | Mechanism investigation. a** CV curves of catalysts collected in hydrogen desorption region in $N_2$-saturated KOH solution. **b** Plot of the area of weak hydrogen adsorption as the index for HBE versus the logarithm of measured exchange current densities. **c** Polarization curves for i-ZnIn-PR/C in $H_2$-saturated 0.1 M KOH and 0.1 M KOD solution. **d** LSV curves of i-ZnIn-PR/C and PR/C for HOR in $H_2$-saturated 0.1 M KOH and 0.1 M KOD solution. Inset: mass activity at 50 mV vs. RH(Deu)E. **e** Scheme for HOR on i-ZnIn-PR/C. In situ DRIFTS spectra of (**f**) Pt/C and (**g**) i-ZnIn-PR/C in 0.1 M KOH of 100 ppm $CO/H_2$. **h** In situ Raman spectra of i-ZnIn-PR/C collected in $H_2$-saturated and $CO/H_2$-saturated (100 ppm) KOH solution (0.1 M). **i** Scheme for resistance to CO-poisoning of i-ZnIn-PR/C.

splitting of Ru-4$d$ significantly increased from 0.68 eV in i-ZnIn-PR to 1.35 eV in PR, leading to increased electron transfer barrier for Ru sites. The site-dependent PDOSs of Pt sites were calculated to further understand the electronic modulations induced by the oxophilic metal atoms (Fig. 6e). From bulk to surface, the overall Pt-5$d$ orbitals gradually upshifted with enhanced electroactivity. However, the introduction of neighboring In and Zn atoms resulted in the slight downshifting of Pt-5$d$ orbitals, especially on the surface sites. This subtle modulation of Pt sites indicated that the oxophilic metal atoms could slightly lower the $d$-band center of the surface Pt site and thus modulate the binding strength with *H during the HOR. For the Ru sites, the $e_g$-$t_{2g}$ splitting of 4$d$ orbitals displayed the alleviation from the bulk to the surface, as a result of improved electroactivity on the surface (Fig. 6f). For surface Ru sites near the Zn or O sites, the $e_g$-$t_{2g}$ splitting was significantly reduced and the electron density near $E_F$ was improved, which facilitated the adsorption of *OH to offer an efficient HOR process. For the doped Zn atoms, a minor upshifting of surface Zn-3$d$ orbital was observed, which benefited the site-to-site electron transfer with Pt and Ru sites through the reduced energy barriers (Fig. 6g). For In, the surface 5$p$

orbital was obviously different from the bulk and middle 5$p$ orbitals (Fig. 6h). The above results suggested that the introduction of In and Zn atoms could strongly modulate the electronic structures of i-ZnIn-PR, which promoted electron transfer and optimized the binding strength of intermediates. Furthermore, we calculated the binding strengths of key intermediates. As shown in Fig. 6i, the HBE on Pt sites was slightly weakened while the OHBE on Ru sites was largely enhanced, leading to enhanced HOR performance. In addition, DFT calculations demonstrated that the adsorption of *CO on i-ZnIn-PR was weaker than that on PR, in line with the strong resistance to CO-poisoning on i-ZnIn-PR. Afterward, the energy barriers for the HOR process were calculated to reveal the mechanism for enhanced performance. It was noted that the dissociation of $H_2$ was the rate-determining step (RDS) for both i-ZnIn-PR and PR (Fig. 6j). The energy barrier of RDS for i-ZnIn-PR was 0.20 eV, much lower than that of PR (0.56 eV), suggesting the superior HOR activity of i-ZnIn-PR to PR. The formation of *$H_2O$ on the i-ZnIn-PR surface shows stronger reaction trends than that of PR, supporting the improved HOR performance. Additionally, the energy barrier for *CO oxidation to $CO_2$ via the intermediate *COOH on i-ZnIn-PR was

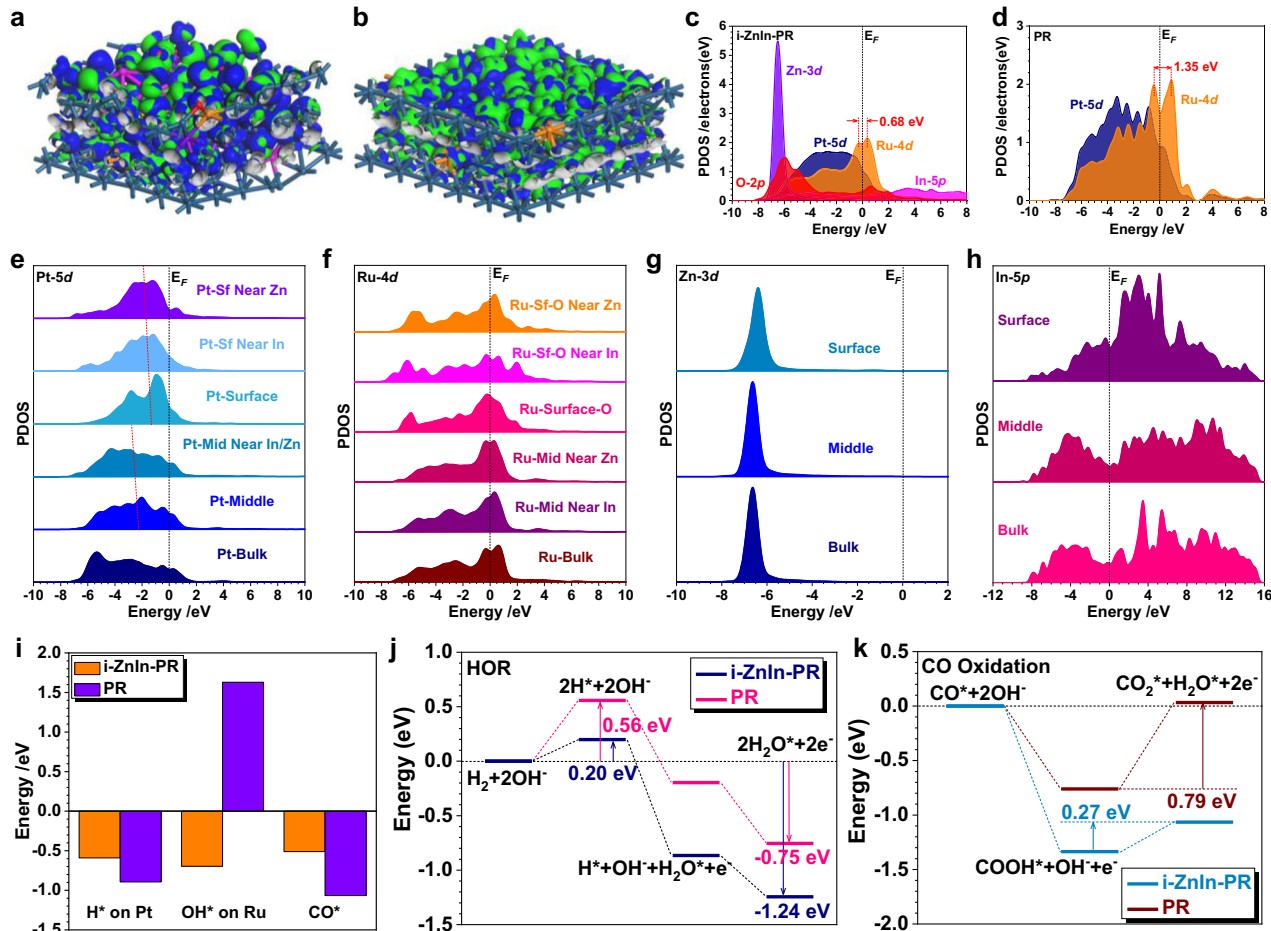

**Fig. 6 | DFT calculations.** 3D contour plot of electronic distributions near the Fermi level of (**a**) i-ZnIn-PR and (**b**) PR. Blue balls: Pt, orange balls: Ru, pink balls: In, purple balls: Zn. Blue and green isosurfaces represent the bonding and anti-bonding orbitals, respectively. PDOSs of (**c**) i-ZnIn-PR and (**d**) PR. The site-dependent PDOSs of (**e**) Pt-5*d*, (**f**) Ru-4*d*, (**g**) Zn-3*d*, and (**h**) In-5*p*. Sf: Surface, Mid: Middle. **i** The HBE, OHBE, and CO adsorption on i-ZnIn-PR and PR. **j** The reaction energies of HOR on i-ZnIn-PR and PR. **k** The reaction energies of CO oxidation on i-ZnIn-PR and PR.

more preferential than that on PR, as evidenced by the lower energy barrier of i-ZnIn-PR (0.27 eV) than that of PR (0.79 eV) (Fig. 6k), which was consistent with the experimental observations.

## Discussion

In this work, we have demonstrated that the implantation of oxophilic metal atoms (e.g., Mn, Fe, Co, Ni, Cu, Zn, Ga, and In) into PR can regulate the electronic properties of Pt and Ru. Combining microstructure characterizations, experimental observations, and theoretical calculations, we have found that the oxophilic metal atoms can interact with Pt and Ru to form Pt−Pt (M) and Ru−O−M coordination (Mn, Fe, Co, Ni, Cu, Zn, Ga, and In), leading to the changes of 4*d* orbitals of Ru and Pt atoms. Detailed investigations revealed that the single state In atoms could promote the HOR activity, while the single state Zn atoms could significantly enhance the stability and resistance to CO-poisoning. When combined dual doping with In and Zn atoms, the strong synergy was able to lower the *d*-band center of the surface Pt site and thus reduced the binding strengths of *H and *CO, while the $e_g$-$t_{2g}$ splitting of 4*d* orbitals for Ru sites reduced and the electron density near Fermi level and facilitated *OH adsorption, as a result of enhanced HOR activity and resistance to CO-poisoning. Consequently, the optimal i-ZnIn-PR/C exhibited a mass activity of 10.2 A mg$_{Pt+Ru}^{-1}$ at 50 mV, which was much higher than that of commercial Pt/C (0.27 A mg$_{Pt}^{-1}$) and PtRu (1.24 A mg$_{Pt+Ru}^{-1}$). An AEMFCs with a low loading (0.1 mg$_{Pt+Ru}$ cm$^{-2}$) of this i-ZnIn-PR/C achieves a PPD and SPD of 1.84 W cm$^{-2}$ and 18.4 W cm$_{Pt+Ru}^{-1}$. Moreover, i-ZnIn-PR/C displayed superior stability to

PtRu/C and Pt/C for alkaline HOR, with a current decay of 5.3% after 10,000 s. Impressively, 84.9% of the current was reserved when i-ZnIn-PR/C was exposed to CO after 5000 s, enabling its practical application for practical HOR. We believe this work not only provides a highly active and stable catalyst for alkaline HOR, but also promotes the fundamental researches on the fabrication and modification of functional materials in catalysis.

## Methods
### Reagents
Platinum (II) acetylacetonate (Pt(acac)$_2$, 97%), ruthenium (III) acetylacetonate (Ru(acac)$_3$, 97%), Nafion perfluorinated resin solution (5 wt.%) and stearyl trimethyl ammonium bromide (STAB, 98%) were purchased from Sigma-Aldrich. Manganese (II) acetylacetonate (Mn(acac)$_2$, 99%) was purchased from Alfa Aesar. Dodecacarbonyl triruthenium (Ru$_3$(CO)$_{12}$, >98%) was supplied by Damas-beta. Zinc carbonate basic (Zn$_2$(OH)$_2$CO$_3$), indium (III) trifluoromethanesulfonate ((CF$_3$SO$_3$)$_3$In, 98%), cupric carbonate basic (Cu$_2$(OH)$_2$CO$_3$, 57%), iron (II) acetylacetonate (Fe(acac)$_2$, 98%), oleylamine (OAm, 80 ~ 90%) and 1-octadecene (ODE, 90%) was purchased from Aladdin. Gallium (III) chloride (GaCl$_3$, 98%) was purchased from TCI-SCT. Nickel(II) carbonate basic tetrahydrate (NiCO$_3$·2Ni(OH)$_2$·4H$_2$O, 44%) was purchased from Shanghai Yuanye Bio-Technology Co., Ltd. Molybdenum hexacarbonyl (Mo(CO)$_6$, 98%) was supplied by Strem Chemicals Inc. Potassium hydrate (KOH, AR), cobalt (II) carbonate hydroxide (2CoCO$_3$·3Co(OH)$_2$, 45%), ethanol (C$_2$H$_6$O, AR), cyclohexane (C$_6$H$_{12}$,

AR) and isopropanol ($C_3H_8O$, AR) were obtained from Sinopharm Chemical Reagent Co., Ltd. Perchloric acid ($HClO_4$, AR) was obtained from Tianjin Zhengcheng Chemical Products Co. Ltd. All the chemicals were used as received without any further purification.

## Preparation of PR nanowires and i-M-PR

For the synthesis of PR nanowires, 10 mg $Pt(acac)_2$, 0.6 mg $Ru(acac)_3$, 42 mg STAB, and 6.2 mg $Mo(CO)_6$ were dissolved in 2.5 mL OAm and 2.5 mL ODE with ultrasonication for 1 h. Afterward, the mixed solution was heated in an oil bath at 200 °C for 5 h and then cooled to room temperature. The products were washed three times with a cyclohexane/ethanol (1/9, v/v) mixture. The synthetic processes for i-M-PR were similar to that for PR nanowires except for adding 1 mg $Ru_3(CO)_{12}$, 4.4 mg $Mo(CO)_6$ and corresponding precursors of M (1.6 mg $(CF_3SO_3)_3In$, 1 mg $Zn_2(OH)_2CO_3$, 0.4 mg $Cu_2(OH)_2CO_3$, 0.7 mg $Fe(acac)_2$, 0.3 mg $NiCO_3\cdot2Ni(OH)_2\cdot4H_2O$, 1 mg $2CoCO_3\cdot3Co(OH)_2$), respectively.

## Characterizations

The morphology of the catalyst was operated on TEM (JEM-1400). HAADF-STEM and HRTEM images were collected on FEI Tecnai F30 TEM at an accelerating voltage of 300 kV. X-ray diffraction patterns were collected on Ultima equipped with a Cu Kα radiation. Metal loading content in the catalysts was determined by inductively coupled plasma optical emission spectrometry (ICP-OES, ICAP 7000, ThermoFisher). XPS measurements were performed on ESCALAB 250 XI (Thermo Scientific, USA). A confocal Raman microscopic spectrometer (IDSpec ARCTIC) was used to record the in situ Raman spectra. Raman spectra were collected from the same surface of the sample at 100% ND filler with 632.8 nm laser excitation. In situ DRIFTS spectra were carried out on a Nicolet 8700 spectrometer (Thermo Scientific) equipped with a nitrogen cooled MCT-A defector using p-polarized light. The XAS spectra were acquired at the TLS01C1 beamline of the National Synchrotron Radiation Research Center (NSRRC, Hsinchu, Taiwan).

## Electrochemical measurements for HOR

Prior to the electrochemical measurements, the prepared nanowires were loaded on commercial carbon support (Vulcan XC-72)[33]. After drying, the catalyst was redispersed in 0.495 mL isopropanol and sonicated for 1 h to obtain a well-mixed catalyst ink. Subsequently, 0.005 mL of Nafion (5 wt.%) was added and sonicated for 1 h to obtain a well-mixed catalyst ink. Finally, the catalyst ink was dropped on a glassy-carbon Rotating Disk Electrode (RDE, geometric area of 0.196 cm²). Note that the loading amounts of noble metals were kept at 2 $\mu g_{Pt}$ in all catalysts for HOR. Electrochemical tests were conducted on a RDE (Pine, diameter: 5 nm) with CHI 760E electrochemical workstation. A saturated calomel electrode and carbon rod electrode were used as the reference electrode and counter electrode, respectively. Prior to the determination of ECSA and HOR, the catalyst was cleaned by CV for 150 cycles in 0.1 M $HClO_4$ solution at a scan rate of 500 mV s⁻¹ and a potential range of −0.25 V to 0.96 V (vs. SCE). After the electrolyte was saturated with pure hydrogen, HOR polarization curves were collected in 0.1 M KOH with iR correction at a rotation rate of 1600 rpm with a sweeping rate of 50 mV s⁻¹. CO tolerance was evaluated in 0.1 M KOH saturated with 1000 ppm $CO/H_2$, while the long-term durability test was evaluated at 0.1 V in $H_2$-saturated KOH (0.1 M). The loading amount of noble metal was kept at 8 $\mu g_{Pt}$. The accelerated durability tests were recorded via cyclic sweeps between −0.1 and 0.4 V (vs. RHE) in 0.1 M KOH electrolyte at a sweep rate of 500 mV s⁻¹ for 2000 cycles. The relevant calculations refer to the previously reported[34]. The CO stripping test was carried out by adsorption in CO-saturated 0.1 M $HClO_4$ for 20 min. Subsequently, the CO stripping curves were obtained by scanning from −0.25 V to 0.96 V (vs SCE) at a scan rate of 50 mV s⁻¹ at room temperature.

The kinetic current ($J_k$) was obtained by the Koutecky–Levich (K-L) equation: $1/J = 1/J_k + 1/J_d = 1/J_k + 1/(Bc_0\omega^{1/2})$, in this equation, $J$ is the measured current, which can be divided into kinetic and diffusional components, $J_k$ is the kinetic current density, $J_d$ is the diffusion-limited current density, B is the Levich constant, $C_0$ is the solubility of $H_2$ in KOH electrolyte, and $\omega$ is the rotating speed during measurements.

The exchange current density ($J_0$) was calculated by fitting $J_k$ into the Butler-Volmer (B-V) equation: $J_k = J_0(e^{\eta\alpha F/RT} - e^{-\eta(1-\alpha)/F/RT})$, in this equation, $\alpha$ is the universal gas constant, F is the Faraday constant (96,485 C mol⁻¹), $\eta$ is the overpotential, R is the universal gas constant, T is the operating temperature, respectively. The B-V equation can be expanded using Taylor's formula and simplified to: $J_k = J_0\eta F/RT$. $J_0$ can be obtained by linear fitting of the polarization curve in the micro-polarization region.

## MEA test

First, the ink was prepared by mixing commercial Pt/C, PtRu/C, and i-ZnIn-PR/C with an ionomer solution composed of the homemade QAPPT dissolved in DMSO (20 wt.% QAPPT, 80 wt% catalysts). The prepared ink was then sprayed on both sides of the alkaline polymer electrolytes (APEs, 25 ± 3 μm in thickness) to produce a catalyst-coated membrane (CCM) with the electrode area of 4 cm⁻²[35]. The obtained CCM was soaked in 1 M KOH solution overnight and then rinsed with deionized water several times to remove the excess KOH. The resulting wet CCM is placed between two pieces of carbon paper to make the membrane electrode assembly (MEA). The $H_2$-$O_2$ fuel cells were tested using an 850E Multi Range fuel cell test station under a galvanic mode at 80 °C. The flow rate of $H_2$ and $O_2$/Air ($CO_2$-free) gases were all 1000 mL min⁻¹ with 0.2 MPa of backpressures. The anode and cathode noble metal loadings were 0.1 mg cm⁻² and 0.4 mg cm⁻², respectively. The prepared CCM was soaked in 1 M KOH solution at 60 °C for overnight, and rinsed with deionized water several times to remove the excess KOH. The resulting wet CCM is placed between two pieces of carbon paper to make the membrane electrode assembly (MEA). The $H_2$-$O_2$ fuel cells were tested using an 850E Multi Range fuel cell test station under a galvanic mode at 80 °C. The flow rate of $H_2$ and $O_2$/Air ($CO_2$-free) gases were all 1000 mL min⁻¹ with 0.2 MPa of backpressures. The anode and cathode of the humidifying temperature were set at 80 °C.

## Computational details

DFT calculations with CASTEP packages were conducted to investigate the electronic modulations induced by the oxophilic metal atoms in i-ZnIn-PR[36]. We have selected the generalized gradient approximation (GGA) with Perdew-Burke-Ernzerhof (PBE) functionals to accurately describe the corresponding exchange-correlation interactions[37–39]. For the plane-wave basis cutoff energy, we have set it to 380 eV based on the default setting of an ultrafine quality. The ultrasoft pseudopotentials with Broyden-Fletcher-Goldfarb-Shannon (BFGS) algorithm have also been applied in this work[40]. Meanwhile, the k-point setting has a separation of 0.71/Å in this work for all the energy minimizations. For all the geometry optimizations, the corresponding convergence needs to satisfy the following criteria: (1) the Hellmann-Feynman forces should be smaller than $1\times10^{-3}$ eV/Å; (2) the total energy difference should be smaller than $5\times10^{-5}$ eV/atom; and (3) the inter-ionic displacement should be less than 0.005 Å. The PtRu has been constructed based on Pt (111) surface with a four-layer thickness in a $5\times5\times1$ supercell. Based on the EDS mapping results, 13% of the Pt has been replaced by Ru atoms, leading to $Pt_{87}Ru_{13}$. The i-ZnIn-PR has been built based on PtRu by introducing 5 In and 5 Zn atoms. Partial oxidation of Ru is also considered by introducing O atoms on Ru sites. We also introduce a 20 Å vacuum space in the z-axis direction to guarantee sufficient space during the relaxation process.

## Data availability

All data supporting the findings of this study are available in the main text or Supplementary Information. Source data in Figs. 1–6 are provided with this paper. Source data are provided with this paper.

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

## Acknowledgements

The authors thank the financial supports by the National Key R&D Program of China (2020YFB1505802, 2021YFA1501101), Ministry of Science and Technology of China (2017YFA0208200), the National Natural

Science Foundation of China (22025108, U21A20327, 22121001), Guangdong Provincial Natural Science Fund for Distinguished Yong Scholars (2021B1515020081), the National Natural Science Foundation of China/Research Grant Council of Hong Kong Joint Research Scheme (N_PolyU502/21), National Natural Science Foundation of China/ Research Grants Council (RGC) of Hong Kong Collaborative Research Scheme (CRS_PolyU504/22), the funding for Projects of Strategic Importance of The Hong Kong Polytechnic University (Project Code: 1-ZE2V), Shenzhen Fundamental Research Scheme-General Program (JCYJ20220531090807017), the Natural Science Foundation of Guang-dong Province (2023A1515012219) and Departmental General Research Fund (Project Code: ZVUL) from The Hong Kong Polytechnic University, and start-up support from Xiamen University. We acknowledge support from the Max Planck-POSTECH-Hsinchu Center for Complex Phase Materials, Research Centre for Carbon-Strategic Catalysis (RC-CSC), Research Institute for Smart Energy (RISE), and Research Institute for Intelligent Wearable Systems (RI-IWEAR) of the Hong Kong Polytechnic University.

## Author contributions

X.H. and Y.X. conceived and supervised the research. Z.L.H., X.H., L.B., and Y.X. designed the experiments. X.H. and Z.L.H. performed most of the experiments and data analysis. S.H. and N.T. performed in situ DRIFTS and in situ Raman test. T.D. performed drawings of the manuscript. Z.L.H., L.L., and M.W. studied the electrochemical performance. Y.H., T.C., and Z.W.H. performed XAS test. Z.L.H., S.L., and J.Z. performed XAS data processing. R.R and L.Z. per-formed the MEA test. M.S. and B.H. performed the DFT calculation. Z.L.H., L.B., Y.X., and X.H wrote the manuscript with support from all co-authors.

## Competing interests

The authors declare no competing interests.

## Additional information

**Supplementary information** The online version contains Supplementary Material available at https://doi.org/10.1038/s41467-024-45369-x.

