## [Peer Review File · Nature Communications]

REVIEWER COMMENTS

Reviewer #1 (Remarks to the Author):

Comments on "Interstitial metal-doped PtRu ultrathin nanowires for hydrogen oxidation catalysis"

In this manuscript, Huang and co-authors reported the incorporation of interstitial metal atoms (Mn, Fe, Co, Ni, Cu, Zn, Ga, and In) with PtRu nanowires (named i-M-PR) can significantly improve the activity and stability of alkaline HOR. Microstructure analysis, X-ray absorption fine structure spectroscopy, cyclic voltammogram (CV), isotopic measurement, and theoretical calculations collectively suggest the dual doping with interstitial In and Zn atoms can weaken *H adsorption, strengthen *OH adsorption, and promote CO conversion to CO₂, as a result of significantly enhanced HOR activity and resistance to CO-poisoning. After all, this piece of research work systematically synthesized PtRu-M nanowires and studied the influence of foreign atoms on the binary alloy performance, which is of interest to the current nanoscience/catalysis societies. It can be published after considering the following comment.

1. One of the primary focuses in this manuscript is the concept of interstitial metal doping, which is also demonstrated in Fig 1a. More compelling evidence is needed to support their claims. The authors may answer how is it possible for elements as large as In, Zn or 3d metal to serve as interstitial metal within an FCC PtRu lattice? If the authors posit the existence of interstitial doping, why do the EXAFS results in Fig 1g, i and Fig 4g, i show a contraction of the Ru-Ru(Pt) and Pt-Pt(Ru) bonds instead? These results seem just the replacements of the Pt or Ru atoms rather than the interstitial metal doping.
2. Due to the inherent CO generation associated with carbon rods, this could significantly impact the study's validity. Authors should use platinum as a counter electrode to bolster the accuracy and reliability of the findings.
3. The authors believe that the HOR performance was strongly dependent on OHBE, with a pivotal step involving the adsorption of *OH species on the catalyst surface (Fig 2e). However, according to the in-situ Raman results (Fig 5h), *OH species on the i-ZnIn-PR/C surface rapidly disappear upon CO introduction. This discrepancy contradicts the author's notion that i-ZnIn-PR/C functions as a CO-tolerant HOR catalyst.
4. As indicated by this manuscript, based on the results from in situ DRIFTS and Raman spectroscopy, the authors concluded that the adsorbed *CO on i-ZnIn-PR/C would combine with *OH to form CO₂ through the intermediate of *COOH. As a result, the i-ZnIn-PR shows excellent CO-tolerant ability. Whether there is a competing reaction between HOR and CO oxidation or a couple reaction involving H₂ and CO? In this part, the roles of Zn or In species should be discussed more in detail.
5. Figure 5i is quite misleading. The author's original statement in the manuscript is '*OH in water' rather than *OH on the catalyst surface. The in-situ Raman results also confirm the disappearance of the crucial intermediate *OH for HOR on the catalyst surface upon CO introduction.
6. In Fig 1d and e, the HRTEM images appear to be fuzzy; therefore, it is recommended to provide the AC HADDF-STEM image for better clarity and resolution. The EDX maps in Figure S3 is not convincing. Please provide better clarity and resolution.

7. In the manuscript, it is worth noting that the ECSA of different catalysts, as determined by the CO-stripping method, may not be considered entirely reliable due to the excellent CO tolerance ability exhibited by these catalysts, other methods such as charges associated with underpotentially deposited H (Hupd) and Cu (Cuupd) might be more suitable.

8. How about the stability of catalysts for a longer continuous operation during HOR? Please provide experimental results

9. For EXAFS analysis

a. One of the parameters should not be the “bond” but the “shell”.

b. It's better to supplement all the fitting parameters, such as amplitude reduction factor S_0^2 .

c. Please check the reasonable and acceptable range of all fitting parameters, for example, the range of σ^2 (\AA^2).

10. Please explain why the i-M-PR nanowires with different interstitial metal atoms (Mn, Fe, Co, Ni, Cu, Zn, Ga, and In) showed different size distributions.

11. As stated in the manuscript, dual doping with interstitial In and Zn atoms can promote CO conversion to CO_2 , as a result of significantly enhanced HOR activity. It seems that the competitive reactions of HOR and CO oxidation take place for i-ZnIn-PR/C. Please explain how to define the kinetically limiting step during HOR and the priority of the reactions. Please provide more detailed evidence.

Reviewer #2 (Remarks to the Author):

In this manuscript, the authors have proposed a universal strategy for introducing metal single atoms into ultrathin PtRu nanowires for enhancing hydrogen oxidation performance in alkaline medium. The slow kinetics and poor stability of hydrogen oxidation in alkaline media is a very challenging problem in the field of electrocatalysis. This reaction has received much attention in recent years due to its importance in anion exchange membrane fuel cells. On the other hand, improving the ability of PtRu catalysts to resist CO poisoning also remains a major challenge. The authors succeeded in realizing the enhancement of hydrogen oxidation activity, stability and resistance to CO-poisoning by introducing metal atoms into PtRu to precisely balance the adsorption strengths for *H , *OH and *CO . Overall, this is an outstanding work. The manuscript could be suitable for publication in Nature Communications after addressing/clarifying several concerns for further improvements.

1. The authors have proved by DFT that bimetallic doping can change the electronic structure of Pt and Ru, but no proof has been given experimentally. XPS analysis of PtRu and i-ZnIn-PR/C catalysts should also be given.

2. How to explain that the HOR kinetics of i-ZnIn-PR/C in 0.1M KOD is slower than that in 0.1M KOH during isotope testing?

3. On the CO poisoning, the author experimentally proved that the introduction of Zn can significantly improve the resistance to CO-poisoning of PtRu. Is it due to the interaction between Zn and Ru?

4. What is the Levich plot on Pt/C, PtRu/C, and i-ZnIn-PR/C?

5. As can be seen from Fig. 5a, the newly developed catalyst shows very special characteristics. Whether OH- adsorption can be observed by scanning CV under N₂-saturated 0.1 M KOH solution?

Reviewer #3 (Remarks to the Author):

In this manuscript, the authors demonstrate that the incorporation of interstitial metal atoms (Mn, Fe, Co, Ni, Cu, Zn, Ga, and In) with PtRu nanowires (named as i-M-PR) can significantly improve the activity and stability for alkaline HOR. Despite many physical/chemical characterizations and theoretical calculations were made in this work, we are afraid that several inconsistencies with previous reports and lack of novelty prevent us from a favorable recommendation of this study for publication. Some specific comments are listed as follows.

1. Enhancing the activity and stability of PtRu for alkaline HOR by incorporation of interstitial metal atoms is a common strategy. According to my knowledge, however, some key scientific findings in this manuscript have been proposed and investigated respectively in previous works including the authors' previous work (Nature communications, 2021, 12, 6261; Angewandte Chemie, 2023: e202217976). All the results in this manuscript can be explained by previous works, and I cannot find any novelty in the manuscript.
2. The author claimed alkaline HOR activity of i-ZnIn-PR/C is higher than that of commercial Pt/C and PtRu/C. However, the intrinsic activity of Pt/C and PtRu/C is just 0.27 A mgPt⁻¹ and 1.24 A mgPt+Ru⁻¹, which is much lower than the currently reported average level.
3. The calculation details and process of relevant parameters should be clearly explained, such as j_0 , j_k and metal loadings. In addition, the values of j_0 s obtained by fitting the micropolarized region need to be compared with that obtained by fitting the Butler-Volmer equation to verify the rationale for the activity assessment. The fitting curve of the Tafel plot and the charge transfer coefficient (α) based on the Butler-Volmer fitting are crucial to verify the accuracy of the activity parameters, which should be provided.
4. Based on the results from in situ DRIFTS and Raman spectroscopy, the authors claimed that the adsorbed *CO on i-ZnIn-PR/C would combine with *OH to form CO₂ through the intermediate of *COOH, as a result of improved resistance to CO-poisoning. This result is very unreasonable and lacks direct evidence, in fact, it is a very difficult process to convert CO into CO₂ through electrochemical methods under the HOR reaction voltage. Therefore, the author's analysis of the improvement of CO anti toxicity is very arbitrary and erroneous.
5. The reaction energies of HOR on i-ZnIn-PR and PR in Fig.6j is very strange, generally, the free energy of final step (2H₂O+2e⁻) should be close to zero
6. The peak position of OH* in Fig 5h of in-situ Raman spectra is quite different from other studies. Besides, deuterium isotopic experiments (D₂O with KOD) should be provided for better comparison.
7. AEMFC tests with the catalysts should be provided and compared with commercial Pt/C and PtRu/C.

Response to Reviewers

Dear Reviewers,

Thanks very much for your precious time to constructive comments on our manuscript titled “**Interstitial metal-doped PtRu ultrathin nanowires for hydrogen oxidation catalysis**” (Manuscript ID: NCOMMS-23-33740-T) for *Nature Communications*. We sincerely appreciate your comments and suggestions on our work, which are highly important for further improvements to our manuscript. According to all the comments, we have made a detailed response and substantial revisions to our revised manuscript.

Reviewer #1 (Remarks to the Author):

In this manuscript, Huang and co-authors reported the incorporation of interstitial metal atoms (Mn, Fe, Co, Ni, Cu, Zn, Ga, and In) with PtRu nanowires (named i-M-PR) can significantly improve the activity and stability of alkaline HOR. Microstructure analysis, X-ray absorption fine structure spectroscopy, cyclic voltammogram (CV), isotopic measurement, and theoretical calculations collectively suggest the dual doping with interstitial In and Zn atoms can weaken *H adsorption, strengthen *OH adsorption, and promote CO conversion to CO₂, as a result of significantly enhanced HOR activity and resistance to CO-poisoning. After all, this piece of research work systematically synthesized PtRu-M nanowires and studied the influence of foreign atoms on the binary alloy performance, which is of interest to the current nanoscience/catalysis societies. It can be published after considering the following comment.

[Authors' Response]: Thanks very much for your kind support of our work. We are very grateful for your interest and appreciation of this work as well as the confirmation of its significance. Based on your valuable comments, we have made the corresponding revisions to improve our manuscript. All the comments are very valuable for us to improve this work, which has been carefully addressed in the revised manuscript. We sincerely hope that the revised manuscript will satisfy your stringent criteria for publication.

1. One of the primary focuses in this manuscript is the concept of interstitial metal doping, which is also demonstrated in Fig 1a. More compelling evidence is needed to support their claims. The authors may answer how is it possible for elements as large as In, Zn or 3d metal to serve as interstitial metal within an FCC PtRu lattice? If the authors posit the existence of interstitial doping, why do the EXAFS results in Fig 1g, i and Fig 4g, i show a contraction of the Ru-Ru(Pt) and Pt-Pt(Ru) bonds instead? These results seem just the replacements of the Pt or Ru atoms rather than the interstitial metal doping.

[Authors' Response]: Thank you very much for this insightful comment. We are sincerely sorry for confusing you and completely agree with you that only those atoms with appropriate size can serve as interstitial metals. Typical for *fcc* PtRu, elements including In and Zn are not typically regarded as typical interstitial elements due to their large atomic sizes. That is, In, Zn, and other 3d transition metals tend to replace Pt/Ru rather than insert the interstice, as commended by you. From XANES and EXAFS results, the presence of Pt-Zn coordinations indicates that In and Zn indeed replace partial Pt atoms. We have revised the relative sentences in the manuscript. Besides, the scheme has been updated to depict the introduction of other atoms into PtRu nanowires (**Fig. 1a**).

Fig. 1 Characterizations of i-M-PR. (a) Scheme for introducing foreign atoms into PR. (b) HAADF-STEM image, (c) XRD pattern. (d) Aberration-corrected HAADF-STEM images, (e) Pt L_{3} -edge XANES spectra, (f) Pt L_{3} -edge FT-EXAFS spectra, (g) Ru K -edge XANES spectra, (h) Ru K -edge FT-EXAFS spectra of i-In-PR. (i) Pt XANES and (j) FT-EXAFS spectra of i-M-PR at L_{3} -edge.

2. Due to the inherent CO generation associated with carbon rods, this could significantly impact the study's validity. Authors should use platinum as a counter electrode to bolster the accuracy and reliability of the findings.

[Authors' Response]: Thank you very much for this valuable comment. The three-electrode configuration, comprising a working electrode (WE), reference electrode (RE), and counter electrode (CE), is a crucial tool in the electrochemical toolkit for investigating the intrinsic activity of electrocatalytic materials. Traditionally, Pt has been widely used as the counter electrode due to its well-known chemical inertness. However, recent reports revealed that Pt might suffer from dissolution in certain electrolytes under oxidative conditions, and even trace amounts of Pt ions will contribute to catalytic performance [ACS Catal. 2020, 10, 10773; ACS Energy Lett. 2017, 2, 1070]. Consequently, carbon-based counter electrodes in recent electrocatalytic studies in alkaline solution, especially for WE with weak CO adsorption [Nat. Catal. 2020, 3, 773]. Furthermore, Ji *et al.* compared Pt and graphite counter electrodes for hydrogen oxidation reaction (HOR) [ACS Catal. 2020, 10, 10773]. No obvious decay of HOR activity has been observed after 50 cycles of potential excursions with both Pt and graphite counter electrodes, suggesting that negligible CO

formed on the graphite counter electrode. Given the remarkable resistance of i-ZnIn-PR/C to CO poisoning and previous reports, we humbly think that the current data are reliable and effective.

3. The authors believe that the HOR performance was strongly dependent on OHBE, with a pivotal step involving the adsorption of *OH species on the catalyst surface (Fig 2e). However, according to the in-situ Raman results (Fig 5h), *OH species on the i-ZnIn-PR/C surface rapidly disappear upon CO introduction. This discrepancy contradicts the author's notion that i-ZnIn-PR/C functions as a CO-tolerant HOR catalyst.

[Authors' Response]: Thank you very much for this insightful comment. It is well known that the HOR activity is determined by the balanced binding strength of H and OH. We initially examined the adsorption ability towards hydrogen by CV curves. It was noted that implantation of oxophilic metal atoms strongly weakens the adsorption ability towards H₂ (Fig. 5a). In addition, the relationship between the weak hydrogen adsorption area and the logarithm of exchange current density shows a positive correlation between hydrogen adsorption and HOR activity (Fig. 5b). Therefore, the introduction of oxophilic metal atoms can weaken hydrogen binding energy (HBE) and promote HOR activity. From the aspect of practical application, the presence of a trace amount of CO in H₂ will strongly poison the HOR catalyst. In this work, we found that the implantation of oxophilic metal atoms (Mn, Fe, Co, Ni, Cu, Zn, Ga, and In) with PtRu nanowires (named as i-M-PR) can significantly improve the activity and stability of alkaline HOR. Detailed investigations show that the introduction of In atoms can promote the HOR activity, while Zn atoms can significantly enhance the stability and resistance to CO-poisoning. In situ diffuse reflectance infrared Fourier transform spectroscopy (DRIFTS) and in situ electrochemical Raman spectroscopy was performed to reveal the mechanism for enhancing the resistance of i-ZnIn-PR/C to CO poisoning. In particular, when commercial Pt/C was exposed to a KOH solution saturated with 100 ppm CO/H₂ (Fig. 5f), the DRIFTS spectra revealed bands at 1384, 1576, and 2910 cm⁻¹ corresponding to *COOH, and a band at 1900 cm⁻¹ could be indexed as bridged-adsorbed CO (B-CO) [ACS Catal. 2022, 12, 14436]. In contrast, no obvious peaks of B-*CO and *COOH were observed in the DRIFTS spectra of i-ZnIn-PR/C after exposure to CO, while the bands at 1538, 1393, and 1306 cm⁻¹ corresponded to CO₃²⁻ (Fig. 5g), indicating that *COOH might be converted into CO₃²⁻ with *OH in water [ACS Catal. 2022, 12, 14436]. Furthermore, prior studies have reported that *OH reacts with *CO in the HOR reactions to eliminate CO [Nat Commun. 2023, 14, 3767; Nano Energy 2017, 34, 22; Nat Commun. 2022, 13, 1596]. In the Raman spectra of i-ZnIn-PR/C, a peak corresponding to *OH appeared at 985 cm⁻¹ in the presence of H₂ at 0.1–0.6 V (left of Fig. 5h), which was further validated by the peak shift from 983 cm⁻¹ to 744 cm⁻¹ when H was replaced by D in isotope experiment (Supplementary Fig. 46). The classical CO electro-oxidation reaction model is the Langmuir-Hinselwood (L–H) mechanism (Langmuir, 2009, 25, 13661), which can be summarized as follows:

Indeed, the introduction of CO led to the weakening or even disappearance of OH peak in the in-situ Raman spectra. However, the limiting current density of i-ZnIn-PR/C at 100 mV (vs. RHE) slightly decreased by ~1.8% after introducing 1000 ppm CO, which was much smaller than those of PR/C (9.2%), commercial PtRu/C (5.4%) and commercial Pt/C (30.0%) (Supplementary Fig. 34). Based on the above results, we concluded that HOR and CO oxidation to CO₂ can

simultaneously occur on i-ZnIn-PR/C surface, the implantation of oxophilic metal atoms (Mn, Fe, Co, Ni, Cu, Zn, Ga, and In) may provide additional sites for CO oxidation.

Fig. 5 Mechanism investigation. **a** CV curves of catalysts collected in hydrogen desorption region in N_2 -saturated KOH solution. **b** Plot of the area of weak hydrogen adsorption as the index for HBE versus the logarithm of measured exchange current densities. **c** Polarization curves for i-ZnIn-PR/C in H_2 -saturated 0.1 M KOH and 0.1 M KOD solution. **d** LSV curves of i-ZnIn-PR/C and PR/C for HOR in H_2 -saturated 0.1 M KOH and 0.1 M KOD solution. Inset: mass activity at 50 mV vs. RH(Deu)E. **e** Scheme for HOR on i-ZnIn-PR/C. In situ DRIFTS spectra of **f** Pt/C and **g** i-ZnIn-PR/C in 0.1 M KOH of 100 ppm CO/H_2 . **h** In situ Raman spectra of i-ZnIn-PR/C collected in H_2 -saturated and CO/H_2 -saturated (100 ppm) KOH solution (0.1 M). **i** Scheme for resistance to CO-poisoning of i-ZnIn-PR/C.

Supplementary Figure 34. Polarization curves of (a) commercial Pt/C, (b) commercial PtRu/C, (c) PR/C, and (d) i-ZnIn-PR/C in 1000 ppm CO/H₂-saturated 0.1 M KOH.

Supplementary Figure 46. In situ Raman spectra of i-ZnIn-PR/C collected in H₂-saturated KOD solution (0.1 M).

4. As indicated by this manuscript, based on the results from in situ DRIFTS and Raman spectroscopy, the authors concluded that the adsorbed *CO on i-ZnIn-PR/C would combine with *OH to form CO₂ through the intermediate of *COOH. As a result, the i-ZnIn-PR shows excellent CO-tolerant ability. Whether there is a competing reaction between HOR and CO oxidation or a couple reaction involving H₂ and CO? In this part, the roles of Zn or In species should be discussed more in detail.

[Authors' Response]: Thank you for this insightful comment. For alkaline HOR, it has been widely discussed that the appropriate adsorption of *H, *OH, and *CO on the electrocatalyst plays

a pivotal role in regulating the activity and stability [Nat. Chem. **2013**, 5, 255; Nat. Chem. **2013**, 5, 255]. Specifically, the HOR activity is determined by the balanced binding strength to H and OH, while the stability is related to the adsorption ability to CO. Ideal catalysts with long-term stability require weak CO adsorption or the capability to convert CO to CO₂. Indeed, HOR and CO electrooxidation can simultaneously occur because composite catalysts generally contain different sites. In our case, PtRu is responsible for HOR, while the introduced Zn and In facilitate the adsorbed CO from PtRu to In and Zn and further oxidation to CO₂. Therefore, i-ZnIn-PR/C demonstrated excellent HOR, AEMFCs properties, and CO toxicity resistance (**Fig. 3**).

Moreover, we used in situ DRIFTS to study the mechanism for CO tolerance. When commercial Pt/C was exposed to KOH solution saturated with 100 ppm CO/H₂ (**Fig. 5f**), the bands at 1384, 1576, and 2910 cm⁻¹ in the DRIFTS spectra of commercial Pt/C were ascribed to *COOH, while the band at 1900 cm⁻¹ could be indexed as bridged-adsorbed CO (B-*CO).²⁷ In contrast, no obvious peaks of B-*CO and *COOH were observed in the DRIFTS spectra of i-ZnIn-PR/C after exposure to CO, while the bands at 1538, 1393, and 1306 cm⁻¹ corresponded to CO₃²⁻ (**Fig. 5g**), indicating that *COOH might be converted into CO₃²⁻ with *OH on the catalyst surface.

In addition, DFT calculations demonstrated that the adsorption of *CO on i-ZnIn-PR was weaker than that on PR, in line with the strong resistance to CO-poisoning on i-ZnIn-PR. The energy barrier for *CO oxidation to CO₂ via the intermediate *COOH on i-ZnIn-PR was more preferential than that on PR, as evidenced by the lower energy barrier of i-ZnIn-PR (0.27 eV) than that of PR (0.79 eV) (**Fig. 6k**), which was consistent with the experimental observations.

Fig 3. HOR performance evaluation of Pt-based catalysts. **a** HOR polarization curves were recorded in H₂-saturated 0.1 M KOH with a sweeping rate of 50 mV s⁻¹ at a rotation rate of 1600 rpm. **b** ECSA normalized specific activity and mass activity at 50 mV (vs. RHE). **c** Comparison of mass activity at 50 mV (vs. RHE) and specific activity between i-ZnIn-PR/C and other reported catalysts. **d** Durability test for i-ZnIn-PR/C and references in an H₂-saturated 0.1 M KOH solution at 100 mV (vs. RHE). **e** CO tolerance test for different catalysts at 100 mV (vs. RHE). **f** Fuel cell performance test of H₂/O₂ AEMFCs with commercial Pt/C (0.1 mg_{Pt} cm⁻²), PtRu/C (0.1 mg_{Pt+Ru} cm⁻²) and i-ZnIn-PR/C (0.1 mg_{Pt+Ru} cm⁻²) in anode and Pt/C (0.4 mg_{Pt} cm⁻²) in cathode. **g** Comparison of the noble metal utilization for H₂-O₂ AEMFC. **h** Fuel cell performance test of H₂/Air (CO₂-free) AEMFC with commercial Pt/C (0.1 mg_{Pt} cm⁻²), PtRu/C (0.1 mg_{Pt+Ru} cm⁻²) and i-ZnIn-PR/C (0.1 mg_{Pt+Ru} cm⁻²) in anode and Pt/C (0.4 mg_{Pt} cm⁻²) in cathode.

Fig 5. Mechanism investigation. In situ DRIFTS spectra of **f** Pt/C and **g** i-ZnIn-PR/C in 0.1 M KOH of 100 ppm CO/H₂.

Fig 6. DFT calculations. **k** The reaction energies of CO oxidation on i-ZnIn-PR and PR.

5. Figure 5i is quite misleading. The author's original statement in the manuscript is '*OH in water' rather than '*OH on the catalyst surface'. The in-situ Raman results also confirm the disappearance of the crucial intermediate '*OH for HOR on the catalyst surface upon CO introduction.

[Authors' Response]: Thank you very much for carefully reviewing our manuscript. As requested, we have updated the **Fig. 5i** in the revised manuscript. Please see the revised sentence of "In

contrast, no obvious peaks of B^*CO and *COOH were observed in the DRIFTS spectra of *i*-ZnIn-PR/C after exposing to CO, while the bands at 1538, 1393 and 1306 cm^{-1} corresponded to CO_3^{2-} (Fig. 5g), indicating that *COOH might be converted into CO_3^{2-} with *OH on the catalyst surface.²⁷” in the revised version.

Fig. 5 i Scheme for resistance to CO-poisoning of *i*-ZnIn-PR/C.

6. In Fig 1d and e, the HRTEM images appear to be fuzzy; therefore, it is recommended to provide the AC HADDF-STEM image for better clarity and resolution. The EDX maps in Figure S3 is not convincing. Please provide better clarity and resolution.

[Authors' Response]: Thanks very much for your kind comments. As requested, we have updated new AC-HADDF-STEM images (**Fig. 1d**) and EDX maps (**Supplementary Figure 3**) in the revised version. The clarity and resolution have both been improved based on your comments.

Fig. 1 d Aberration-corrected HAADF-STEM images of *i*-In-PR.

Supplementary Figure 3 (a) HAADF-STEM image with elemental mappings of *i*-In-PR.

7. In the manuscript, it is worth noting that the ECSA of different catalysts, as determined by the CO-stripping method, may not be considered entirely reliable due to the excellent CO tolerance ability exhibited by these catalysts, other methods such as charges associated with underpotentially deposited H (H_{upd}) and Cu (Cu_{upd}) might be more suitable.

[Authors' Response]: Thanks very much for this valuable comment. As requested, we carried out the underpotential deposition H (H_{upd}) and Cu (Cu_{upd}) experiments with i-ZnIn-PR/C. As shown in **Supplementary Figure 19**, the ECSA of i-ZnIn-PR/C measured by H_{upd} was $66.7 \text{ m}^2 \text{ g}_{\text{Pt+Ru}}^{-1}$, which was smaller than the ECSA obtained by CO stripping ($77.8 \text{ m}^2 \text{ g}_{\text{Pt+Ru}}^{-1}$). Note that ECSA was underestimated because the introduction of In and Zn weakened the adsorption of hydrogen by the catalyst (**Fig. 5a, b**). Moreover, we also tested the ECSA of i-ZnIn-PR/C by the Cu_{upd} method, which was close to that obtained by CO stripping (**Supplementary Figure 19d**). The above results suggest that the CO stripping experiment used in this study is reasonable for calculating ECSA.

Supplementary Figure 19 (a) Hydrogen adsorption/desorption for measuring the surface area of i-ZnIn-PR/C at 50 mV s^{-1} in 0.1 M HClO_4 . (b) CO stripping quantifies the surface area of i-ZnIn-PR/C at 50 mV s^{-1} in 0.1 M HClO_4 . (c) Underpotential deposition of Cu for measuring the surface area of i-ZnIn-PR/C at 50 mV s^{-1} in 0.5 M HClO_4 in the presence of 5 mM CuSO_4 . (d) ECSAs of i-M-PR/C at 50 mV (vs. RHE).

8. How about the stability of catalysts for a longer continuous operation during HOR? Please provide experimental results.

[Authors' Response]: Thank you very much for this insightful comment. As requested, the stability of catalysts for a longer continuous operation is given in **Supplementary Figure 21**. It is noted that 99.3% of the initial current was reserved for i-Zn-PR/C at 4000 s at 100 mV in the durability test, which was much higher than those of other references.

Supplementary Figure 21 Relative current-time chronoamperometry of different i-M-PR/C in H₂-saturated 0.1 M KOH solution at an overpotential of 100 mV vs. RHE.

9. For EXAFS analysis

a. One of the parameters should not be the “bond” but the “shell”.

[Authors' Response]: Thanks very much for your kind comments. As requested, we have revised “bond” to “shell” in the revised version.

b. It's better to supplement all the fitting parameters, such as amplitude reduction factor S_0^2 .

[Authors' Response]: Thanks very much for your kind comments. As requested, we have added the amplitude reduction factor S_0^2 and the relevant note in our revised manuscript for **Supplementary Table 8-11**.

c. Please check the reasonable and acceptable range of all fitting parameters, for example, the range of σ^2 (\AA^2).

[Authors' Response]: Thanks very much for your kind comments. All the fitting parameters have been checked as supplied in **Supplementary Table 8-11**.

Supplementary Table 8. Structural parameters of Zn foil, ZnO, and i-ZnIn-PR extracted from the EXAFS fitting.

Sample	Shell	CN	R (Å)	$\sigma^2/\text{Å}^2$	R-factor
Zn foil	Zn–Zn	6	2.6 ± 0.02	0.0130 ± 0.0011	0.019
ZnO	Zn–O	4	2.0 ± 0.03	0.0093 ± 0.0024	0.017
i-ZnIn-PR	Zn–O	3.02 ± 0.22	2.06 ± 0.03	0.0125	0.019
i-ZnIn-PR	Zn–Pt	1.61 ± 0.37	2.66 ± 0.04	0.0077	0.019

Note: CN is the coordination number; R is interatomic distance; σ^2 is the Debye-Waller factor (a measure of thermal and static disorder in absorber-scatterer distances); S_0^2 for Zn-Zn was set as 0.70, which is obtained from the experimental EXAFS fit of the Zn foil reference by fixing CN to the known crystallographic value and was fixed to all samples; R-factor is used to value the goodness of the fitting.

Supplementary Table 9. Structural parameters of In₂O₃ and i-ZnIn-PR extracted from the EXAFS fitting.

Sample	Shell	CN	R (Å)	$\sigma^2/\text{Å}^2$	R-factor
In ₂ O ₃	In–O	2	2.17 ± 0.01	0.0059 ± 0.0016	0.004
i-ZnIn-PR	In–O	1.67 ± 0.32	2.15 ± 0.02	0.0055 ± 0.0028	0.018

Note: CN is the coordination number; R is interatomic distance; σ^2 is the Debye-Waller factor (a measure of thermal and static disorder in absorber-scatterer distances); S_0^2 for In-O was set as 0.99, which is obtained from the experimental EXAFS fit of the In₂O₃ reference by fixing CN to the known crystallographic value and was fixed to all samples; R-factor is used to value the goodness of the fitting.

Supplementary Table 10. Structural parameters of Pt foil and i-ZnIn-PR extracted from the EXAFS fitting.

Sample	Shell	CN	R (Å)	$\sigma^2/\text{Å}^2$	R-factor
Pt foil	Pt–Pt	12	2.8 ± 0.01	0.0050 ± 0.0011	0.013
i-ZnIn-PR	Pt–Pt (M)	6.5 ± 1.4	2.7 ± 0.02	0.0066	0.019

Note: CN is the coordination number; R is interatomic distance; σ^2 is the Debye-Waller factor (a measure of thermal and static disorder in absorber-scatterer distances); S_0^2 for Pt-Pt was set as 0.84, which is obtained from the experimental EXAFS fit of the Pt foil reference by fixing CN to the known crystallographic value and was fixed to all samples; R-factor is used to value the goodness of the fitting.

Supplementary Table 11. Structural parameters of Ru foil and i-ZnIn-PR extracted from the EXAFS fitting.

Sample	Shell	CN	R (Å)	$\sigma^2/\text{Å}^2$	R-factor
--------	-------	----	-------	-----------------------	----------

Ru foil	Ru–Ru	12	2.7 ± 0.01	0.0026 ± 0.0013	0.021
i-ZnIn- PR i-	Ru–O	3.8 ± 1.43	2.1 ± 0.02	0.0100 ± 0.0036	0.016
ZnIn- PR	Ru–Ru (M)	5.2 ± 1.4	2.7 ± 0.03	0.0181 ± 0.0040	0.016

Note: CN is the coordination number; R is interatomic distance; σ^2 is the Debye-Waller factor (a measure of thermal and static disorder in absorber-scatterer distances); S_0^2 for Ru-Ru was set as 0.64, which is obtained from the experimental EXAFS fit of the Ru foil reference by fixing CN to the known crystallographic value and was fixed to all samples; R-factor is used to value the goodness of the fitting.

10. Please explain why the i-M-PR nanowires with different interstitial metal atoms (Mn, Fe, Co, Ni, Cu, Zn, Ga, and In) showed different size distributions.

[Authors' Response]: Thanks very much for your kind comments. We have calculated the diameter and length distributions of i-M-PR, and the results show that the diameter distributions of i-M-PR are close, with small differences in length (**Supplementary Figure 13**). However, the growth of nanostructures is quite complicated, and the final dimensions are determined by both thermodynamics and kinetics. In our case, oleylamine and 1-octadecene were used as the solvent and reductant for synthesizing nanowires, respectively. The introduction of precursors of M will influence the reduction kinetics. In other words, the reduction kinetics of $(CF_3SO_3)_3In$, $Zn_2(OH)_2CO_3$, $Cu_2(OH)_2CO_3$, $Fe(acac)_2$, $NiCO_3 \cdot 2Ni(OH)_2 \cdot 4H_2O$, $2CoCO_3 \cdot 3Co(OH)_2$ are different in the mixture of oleylamine and 1-octadecene. In a typical mechanism for the growth of nanostructure, the final size is strongly related to the kinetics during nucleation and growth. The faster rate for nucleation means more nucleus are formed during nucleation, leading to the small size. On the contrary, slow kinetics will lead to a low amount of nucleus, and the dissociative ions will overgrow on the nucleus to form large nanoparticles. In this work, $(CF_3SO_3)_3In$, $Zn_2(OH)_2CO_3$, $Cu_2(OH)_2CO_3$, $Fe(acac)_2$, $NiCO_3 \cdot 2Ni(OH)_2 \cdot 4H_2O$, $2CoCO_3 \cdot 3Co(OH)_2$ were used as the precursors of M, and their reduction potentials are different, which leads to the difference in reduction kinetics as well as the length of nanowires.

Supplementary Figure 13. Summary of distributions of diameters and lengths for i-M-PR.

11. As stated in the manuscript, dual doping with interstitial In and Zn atoms can promote CO conversion to CO₂, as a result of significantly enhanced HOR activity. It seems that the competitive reactions of HOR and CO oxidation take place for i-ZnIn-PR/C. Please explain how to define the kinetically limiting step during HOR and the priority of the reactions. Please provide more detailed evidence.

[Authors' Response]: Thanks very much for your kind comments. As we mentioned in the introduction section, H₂ produced via a conventional reforming process may inevitably contain a trace amount of CO, which will poison the catalyst for HOR due to the strong adsorption. Note that HOR performance was tested in pure H₂ without CO. Our experimental and theoretical results show that the introduction of In and Zn can strongly facilitate CO oxidation to CO₂ (**Fig. 5f, 5g, and Fig. 6k**). We then tested the HOR performance in the presence of CO. The limiting current density of i-ZnIn-PR/C at 100 mV (vs. RHE) slightly decreased by ~1.8% after introducing 1000 ppm CO, which was much smaller than those of PR/C (9.2%), commercial PtRu/C (5.4%) and commercial Pt/C (30.0%) (**Supplementary Fig. 34**). Besides, the currents decreased by 15.1% for i-ZnIn-PR/C after 5000 s, while the currents strongly decreased by 43.6% and 52.4% for PR/C and commercial PtRu/C, respectively, after only 4000 s, indicating the superior resistance to CO-poisoning of i-ZnIn-PR/C to other references (**Fig. 3e**).

To further reveal the enhanced HOR performance over i-ZnIn-PR/C, we separately introduced Zn and In into PR, which were named as i-ZnPR/C and i-In-PR/C, respectively. It was noted that the introduction In can promote the HOR activity (**Fig. 2d**), while Zn facilitates the resistance to CO poisoning (**Supplementary Fig. 22**). Given the superior HOR activity and CO tolerance, we concluded that Zn atoms potentially provide additional sites for CO oxidation. In contrast, in the absence of Zn, CO will strongly adsorb on PtRu, as a result of serious decay of HOR activity. It has been reported that the Tafel slope is the rate-determining step (RDS) with a Tafel slope of ~30 mV dec⁻¹, while Volmer or Heyrovsky step is the RDS with a Tafel slope of ~120 mV dec⁻¹ [*J. Phys. Chem. C*, **2015**, 119, 13481]. The Tafel slope of i-ZnIn-PR/C is ~29.6 mV dec⁻¹ (**Fig. R1**), indicating that the Tafel step is the RDS for HOR.

Fig 5. Mechanism investigation. In situ DRIFTS spectra of **f** Pt/C and **g** i-ZnIn-PR/C in 0.1 M KOH of 100 ppm CO/H₂.

Fig 6. DFT calculations. k The reaction energies of CO oxidation on i-ZnIn-PR and PR.

Fig 3. f CO tolerance test for different catalysts at 100 mV (vs. RHE).

Fig 2. d ECSA normalized specific activity and mass activity at 50 mV (vs. RHE).

Supplementary Figure 21. Relative current-time chronoamperometry of different i-M-PR/C in 1000 ppm CO 0.1 M KOH solution at an overpotential of 100 mV vs. RHE.

Fig. R1 Tafel slope of HOR branch for i-ZnIn-PR/C in H₂-saturated 0.1 M KOH solution.

Reviewer #2 (Remarks to the Author):

In this manuscript, the authors have proposed a universal strategy for introducing metal single atoms into ultrathin PtRu nanowires for enhancing hydrogen oxidation performance in alkaline medium. The slow kinetics and poor stability of hydrogen oxidation in alkaline media is a very challenging problem in the field of electrocatalysis. This reaction has received much attention in recent years due to its importance in anion exchange membrane fuel cells. On the other hand, improving the ability of PtRu catalysts to resist CO poisoning also remains a major challenge. The authors succeeded in realizing the enhancement of hydrogen oxidation activity, stability and resistance to CO-poisoning by introducing metal atoms into PtRu to precisely balance the adsorption strengths for *H, *OH and *CO. Overall, this is an outstanding work. The manuscript could be suitable for publication in Nature Communications after addressing/clarifying several concerns for further improvements.

[Authors' Response]: Thanks very much for your kind support and recognition of our work. We highly appreciate your precious time and great efforts in reviewing our work. We are very grateful for your support in this work as well as your constructive comments. Your interest and detailed comments on our work are a great encouragement for our research as well as future works. Based on your comments, we have carefully supplied detailed explanations and further experimental and theoretical data to support our work. We sincerely hope that our revised manuscript will satisfy your high requirements for publication.

1. The authors have proved by DFT that bimetallic doping can change the electronic structure of Pt and Ru, but no proof has been given experimentally. XPS analysis of PtRu and i-ZnIn-PR/C catalysts should also be given.

[Authors' Response]: Thanks very much for your kind comments. As requested, we have collected the XPS spectra of PR/C and i-ZnIn-PR/C (**Supplementary Fig. 44**). Compared to PR/C, the Pt 4f XPS spectrum of i-ZnIn-PR/C shifts to lower energy, while the peak of Ru 3p XPS spectrum shifts to higher energy, indicating that electrons may be transferred from Ru to Pt in i-ZnIn-PR/C. Moreover, the d-band center of i-ZnIn-PR/C shifts downward compared with PR/C (3.71 eV), leading to the weakened adsorption of H (**Supplementary Fig. 45**).

Supplementary Fig. 44. (a) Pt 4f and (b) Ru 3p XPS spectra of PR/C and i-ZnIn-PR/C.

Supplementary Fig. 45. d-band center of PR/C and i-ZnIn-PR/C.

2. How to explain that the HOR kinetics of i-ZnIn-PR/C in 0.1M KOD is slower than that in 0.1M KOH during isotope testing?

[Authors' Response]: Thanks very much for your kind comments. The ionic product K_w of D₂O (pK_w) was 14.951, which was higher than that of H₂O ($pK_w = 13.995$), as a result of a higher pD of 13.7 in 0.1 M KOD than that in 0.1 M KOH (12.8) [*Nat. Mater.* **2022**, 21, 804; *J. Inorg. Biochem.* **2004**, 98, 161]. Consequently, the polarization curve of i-ZnIn-PR/C collected in KOD shifts to low potential compared to that collected in KOH (**Fig. 5c**).

3. On the CO poisoning, the author experimentally proved that the introduction of Zn can significantly improve the resistance to CO-poisoning of PtRu. Is it due to the interaction between Zn and Ru?

[Authors' Response]: Thanks very much for your kind comments. Ru and Zn of the catalyst play a crucial role in the catalytic reaction. According to the corresponding EXAFS spectrum analysis, the absence of Pt–O coordination but the presence of Ru–O coordination in corresponding EXAFS spectra (**Fig. 4g, i**), while Zn–O coordination exists in the EXAFS spectrum of Zn K-edge (**Fig. 4k**), indicating that Zn single atoms are coordinated with O in the form of Zn–O–Ru. Additionally, DFT calculation shows that the energy barrier for ^{*}CO oxidation to CO₂ via the intermediate ^{*}COOH on i-ZnIn-PR was more preferential than that on PR, as evidenced by the lower energy barrier of i-ZnIn-PR (0.27 eV) than that of PR (0.79 eV) (**Fig. 6k**). At the same time, the CO toxicity resistance of i-Zn-PR/C is stronger than that of i-In-PR/C (**Fig. 2f**), indicating that there is a strong interaction between Zn and Ru, and thus improve the resistance to CO-poisoning.

4. What is the Levich plot on Pt/C, PtRu/C, and i-ZnIn-PR/C?

[Authors' Response]: Thanks very much for your kind comments. As requested, the Koutecky–Levich plots of Pt/C, PtRu/C, and i-ZnIn-PR/C were given in **Supplementary Fig. 27**. The corresponding updated discussions are supplied below.

“Then, we collected the HOR polarization curves of Pt/C, PtRu/C, and i-ZnIn-PR/C as a function of rotation rate, where the current density increased with rotation rate due to the promotion on mass transport (Supplementary Fig. 27). The slopes for i-ZnIn-PR/C ($12.6 \text{ mA}^{-1} \text{ cm}^{-2} \text{ s}^{1/2}$) is close to the theoretical value (Supplementary Fig. 27d), further confirming the H_2 mass transport control process.²³” in the revised manuscript.

Supplementary Fig. 27. Polarization curves of (a) commercial Pt/C, (b) commercial PtRu/C, and (c) i-ZnIn-PR/C at different various rotation speeds, respectively. (d) Koutecky–Levich plots of commercial Pt/C, commercial PtRu/C, and i-ZnIn-PR/C at an overpotential of 100 mV.

5. As can be seen from Fig. 5a, the newly developed catalyst shows very special characteristics. Whether OH^- adsorption can be observed by scanning CV under N_2 -saturated 0.1 M KOH solution?

[Authors' Response]: Thanks very much for your kind comments. As requested, we have added the full CV of Pt/C, PR/C, and i-ZnIn-PR/C (**Fig. R2**). It is noted that i-ZnIn-PR/C manifests lower OH^- adsorption (0.804 V) potential than commercial Pt/C (0.861 V), indicating that the surface of i-ZnIn-PR/C prefers to bind OH^- compared to that of commercial Pt/C [*Nat Commun.* **2023**, 14, 3767].

Fig. R2. CV curves of catalysts collected in N₂-saturated KOH solution.

Reviewer #3 (Remarks to the Author):

In this manuscript, the authors demonstrate that the incorporation of interstitial metal atoms (Mn, Fe, Co, Ni, Cu, Zn, Ga, and In) with PtRu nanowires (named as i-M-PR) can significantly improve the activity and stability for alkaline HOR. Despite many physical/chemical characterizations and theoretical calculations were made in this work, we are afraid that several inconsistencies with previous reports and lack of novelty prevent us from a favorable recommendation of this study for publication. Some specific comments are listed as follows.

[Authors' Response]: Thanks very much for your kind support of our work. We highly appreciate your time and efforts in reviewing our work. All your comments are very constructive for improving our research. Based on your valuable comments, we have made the corresponding revisions and supplied additional explanations to address all the concerns. We sincerely hope that the revised manuscript will satisfy your stringent criteria for publication.

1. Enhancing the activity and stability of PtRu for alkaline HOR by incorporation of interstitial metal atoms is a common strategy. According to my knowledge, however, some key scientific findings in this manuscript have been proposed and investigated respectively in previous works including the authors' previous work (Nature communications, 2021, 12, 6261; Angewandte Chemie, 2023: e202217976). All the results in this manuscript can be explained by previous works, and I cannot find any novelty in the manuscript.

[Authors' Response]: Thanks very much for your kind comments. It is well known that the highly efficient HOR process is strongly impeded by the current disadvantages including unsatisfying stability and activity of electrocatalysts, especially in alkaline conditions. Moreover, the trace amount of CO in H₂ produced through the conventional reforming process will poison the active sites of noble metal-based catalysts. In principle, an ideal HOR catalyst should be capable of balanced binding strengths to *H and *OH, as well as improved resistance to CO poisoning, which is formidably challenging yet. Over the past decades, substantial efforts have been devoted to increasing the activity and stability of catalysts for HOR.

As you commended, introducing additional elements into Pt or Ru has been regarded as one of the most important strategies for enhancing HOR performance. Despite significant progress has been achieved, great challenges still remain for realizing the HOR process with high efficiency. In this work, we have proposed **a universal strategy for implanting single In and Zn atoms into**

ultrathin PtRu nanowires (noted as i-ZnIn-PR) for enhancing HOR performance. Detailed investigations show that the introduction of In atoms can promote the HOR activity, while Zn atoms can significantly enhance the stability and resistance to CO-poisoning. When combined dual doping with In and Zn atoms, the strong synergy can lower the *d*-band center of the surface Pt site and thus reduce the binding strengths of ^{*}H and ^{*}CO, while the *e_g-t_{2g}* splitting of *4d* orbitals for Ru sites reduces and the electron density near Fermi level and facilitates ^{*}OH adsorption, as a result of enhanced HOR activity and CO tolerance. Impressively, the mass activity of i-ZnIn-PR/C reaches 10.2 A mg_{Pt+Ru}⁻¹ at 50 mV, which is 1–2 orders of magnitude higher than that of commercial Pt/C and PtRu/C. Moreover, i-ZnIn-PR/C exhibits superior stability at 100 mV vs. RHE with a slight current decay of 5.3% after 10000 s. Impressively, i-ZnIn-PR/C displays superior CO tolerance to commercial PtRu/C and Pt/C, where 94.7% of the initial current is reserved after 10000 s at 100 mV vs. RHE and 84.9% of the initial current was maintained in the presence of 1000 ppm CO after 5000 s, respectively. More importantly, the peak power density and specific power density are as high as 1.84 W cm⁻² and 18.4 W mg_{Pt+Ru}⁻¹ with a low loading (0.1 mg cm⁻²) of this catalyst.

In summary, the novelty of this work is that we have proposed a **universal strategy for introducing single atoms including Mn, Fe, Co, Ni, Cu, Zn, Ga, and In into ultrathin PtRu nanowires**. Screening experiments show that In atoms can promote HOR activity, and Zn atoms can facilitate CO oxidation to CO₂, as a result of significantly enhanced activity and resistance to CO poisoning. Therefore, we think that this work will supply important contributions to the design of advanced and novel HOR electrocatalysts in the future.

2. The author claimed alkaline HOR activity of i-ZnIn-PR/C is higher than that of commercial Pt/C and PtRu/C. However, the intrinsic activity of Pt/C and PtRu/C is just 0.27 A mg_{Pt}⁻¹ and 1.24 A mg_{Pt+Ru}⁻¹, which is much lower than the currently reported average level.

[Authors' Response]: Thanks very much for your kind comments. We have re-tested commercial Pt/C and PtRu/C for HOR three times (**Fig. R3 and Fig. R44**). For commercial Pt/C, the mass activity was 0.26 A mg_{Pt}⁻¹, 0.27 A mg_{Pt}⁻¹, and 0.32 A mg_{Pt}⁻¹, respectively, for each single test. Moreover, the mass activity values obtained from three repeated tests were 1.20 A mg_{Pt+Ru}⁻¹, 1.29 A mg_{Pt+Ru}⁻¹, and 1.32 A mg_{Pt+Ru}⁻¹, respectively, for each single test. Our collected data of Pt/C and PtRu/C were also close to those in previous reports (**Table R1**).

Fig. R3 HOR polarization curves of commercial (a) Pt/C-1, (b) Pt/C-2, and (c) Pt/C-3 in H₂-saturated 0.1 M KOH solutions with a rotation speed of 1600 rpm. (d) Normalized mass activity at an overpotential of 50 mV vs. RHE for Pt/C-1, Pt/C-2, and Pt/C-3. (Pt/C-R, R stands for number of repetitions).

Fig. R4 HOR polarization curves of commercial (a) PtRu/C-1, (b) PtRu/C-2, and (c) PtRu/C-3 in H₂-saturated 0.1 M KOH solutions with the rotation speed of 1600 rpm. (d) Normalized mass activity at an overpotential of 50 mV vs. RHE for PtRu/C-1, PtRu/C-2, and PtRu/C-3. (PtRu/C-R, R stands for number of repetitions).

Table R1 Summary of the reported commercial Pt/C and PtRu/C catalysts for mass activity.

Catalyst	$J_m, 50mV$ (A mg _{Pt} ⁻¹)	$J_m, 50mV$ (A mg _{Pt+Ru} ⁻¹)	Ref
Pt/C	0.27	0.27	This work
PtRu/C	3.7	1.24	This work
Pt/C	0.31	0.31	Adv. Mater. 2023, 2211854.
Pt/C	0.28	0.28	J. Mater. Chem. A , 2022,10, 18972
Pt/C	0.34	0.34	Nat. Commun. 2021, 12, 6261.
Pt/C	0.31	0.31	Adv. Mater. 2023, 2211854.
Pt/C	0.145	0.145	Adv. Energy Mater. 2022, 12, 2103336.
Pt/C	0.36	0.36	Small 2021, 17, 2006698.
Pt/C	0.29	0.29	Matter 2023, 6, 193.
PtRu/C	2.51	1.65	Nat. Commun. 2021, 12, 6261.
PtRu/C		1.4	Matter 2023, 6, 193.

3. The calculation details and process of relevant parameters should be clearly explained, such as j_0 , j_k and metal loadings. In addition, the values of $j_{0,s}$ obtained by fitting the micropolarized region need to be compared with that obtained by fitting the Butler-Volmer equation to verify the rationale for the activity assessment. The fitting curve of the Tafel plot and the charge transfer coefficient (α) based on the Butler-Volmer fitting are crucial to verify the accuracy of the activity parameters, which should be provided.

[Authors' Response]: Thanks very much for your kind comments. As requested, we have incorporated detailed calculations and processes for hydrogen oxidation reaction (HOR) into the

revised manuscript. The relevant results have been included for clarity and comprehensibility, which are listed below. Additionally, the quantification of the metal load was carried out using Inductively Coupled Plasma (ICP) analysis.

“The kinetic current (J_k) was obtained by the Koutecky–Levich (K-L) equation: $1/J = 1/J_k + 1/J_d = 1/J_k + 1/(Bc_0\omega^{1/2})$, in this equation, J is the measured current, which can be divided into kinetic and diffusional components, J_k is the kinetic current density, J_d is the diffusion-limited current density, B is the Levich constant, C_0 is the solubility of H_2 in KOH electrolyte, and ω is the rotating speed during measurements. The exchange current density (J_0) was calculated by fitting J_k into the Butler-Volmer (B-V) equation: $J_k = J_0(e^{\eta\alpha F/RT} - e^{-\eta(1-\alpha)F/RT})$, in this equation, α is the universal gas constant, F is the Faraday constant ($96,485 \text{ C mol}^{-1}$), η is the overpotential, R is the universal gas constant, T is the operating temperature, respectively. The B-V equation can be expanded using Taylor's formula and simplified to $J_k = J_0\eta F/RT$. J_0 can be obtained by linear fitting of the polarization curve in the micro-polarization region.”

Furthermore, we have calculated the area-specific exchange current density ($J_{0,s}$) by normalization of the $J_{0,m}$. By linear fitting the micro-polarized regions of HOR (**Supplementary Figure 29**), the $J_{0,s}$ of i-ZnIn-PR/C is 2.15 mA cm^{-2} , which is higher than that of Pt/C (0.24 mA cm^{-2}), PtRu/C (0.49 mA cm^{-2}), and PR/C (0.63 mA cm^{-2}) (**Table R2**). By fitting the Butler-Volmer equation, we determined a $J_{0,s}$ of 1.97 mA cm^{-2} for i-ZnIn-PR/C, higher than those for Pt/C (0.22 mA cm^{-2}), PtRu/C (0.48 mA cm^{-2}), and PR/C (0.61 mA cm^{-2}), which are close to the corresponding values obtained by the micro-polarized regions fitting. These results indicate that i-ZnIn-PR/C exhibits faster HOR dynamics, compared to Pt/C, PtRu/C, and PR/C. Subsequently, we conducted Butler-Volmer fitting on the polarization curves for the catalysts during the HER/HOR, and the results are shown in **Supplementary Figure 30**. Additionally, we have included the charge transfer coefficient (α) obtained through the Butler-Volmer fitting in **Supplementary Table 4**. The corresponding updated discussions are supplied below.

“The exchange currents (I_0) and area specific exchange current density ($J_{0,s}$) by fitting with Butler-Volmer (BV) equation and micro-polarization region (MR) further confirmed the superior HOR activity of i-ZnIn-PR/C to other references (Supplementary Fig. 28, 29 and Supplementary Table 4). The fitted curve of the Tafel plot and the charge transfer coefficient (α) extrapolated based on the Butler-Volmer fitting are shown in Supplementary Fig. 30 and Supplementary Table 4.” in the revised manuscript.

Supplementary Figure 29. Fitting the micro-polarization regions of HOR of Pt/C, PtRu/C, PR/C, and i-ZnIn-PR/C.

Table R2 Comparison of $J_{0,s}$ values obtained by fitting the micro-polarization regions and the Butler-Volmer equation.

Catalyst	$J_{0,s}/\text{mA cm}^{-2}$ (BV)	$J_{0,s}/\text{mA cm}^{-2}$ (MR)
Pt/C	0.22	0.24
PtRu/C	0.48	0.49
PR/C	0.61	0.63
i-ZnIn-PR/C	1.97	2.15

Supplementary Figure 30. Butler-Volmer fitting on the HER/HOR polarization curves for (a) Pt/C, (b) PtRu/C, (c) PR/C, (D) i-ZnIn-PR/C.

Supplementary Table 4. The specific activities, mass activities, exchange current, and charge transfer coefficient for i-M-PR/C.

Catalyst	$J_m, 50\text{mV}$ (A mgPt^{-1})	$J_m, 50\text{mV}$ (A mgPt+Ru^{-1})	$J_s, 50\text{mV}$ (mA cm^{-2})	I_0 (mA)	α
Pt/C	0.27	0.27	0.48	0.24	0.60
PtRu/C	3.7	1.24	1.80	0.66	0.77
PR/C	1.84	1.60	2.50	0.90	0.75
i-Mn-PR/C	6.70	5.77	8.96	1.68	0.95
i-Fe-PR/C	6.77	5.74	8.17	1.74	0.99
i-Co-PR/C	4.32	3.73	5.26	2.08	0.95
i-Ni-PR/C	1.98	1.69	3.29	0.79	0.78
i-Cu-PR/C	2.33	2.01	3.30	0.85	0.82

i-Zn-PR/C	5.37	4.57	6.72	2.09	0.87
i-Ga-PR/C	3.32	2.87	3.89	1.64	0.81
i-In ₂ -PR/C	8.53	7.14	11.86	2.28	1.01
i-In _{3.9} -PR/C	9.96	8.51	12.21	3.01	0.99
i-In _{5.1} -PR/C	6.77	5.72	7.74	1.46	0.99
i-In _{6.3} -PR/C	3.03	2.50	3.46	1.38	0.79
i-ZnIn-PR/C	11.7	10.2	14.77	3.53	1.02

4. Based on the results from in situ DRIFTS and Raman spectroscopy, the authors claimed that the absorbed *CO on i-ZnIn-PR/C would combine with *OH to form CO₂ through the intermediate of *COOH, as a result of improved resistance to CO-poisoning. This result is very unreasonable and lacks direct evidence, in fact, it is a very difficult process to convert CO into CO₂ through electrochemical methods under the HOR reaction voltage. Therefore, the author's analysis of the improvement of CO anti toxicity is very arbitrary and erroneous.

[Authors' Response]: Thanks very much for your kind comments. Currently, to improve the toxic resistance of Pt catalyst, two schemes can be adopted. (1) **Electronic effect:** This approach involves alloying a second element with Pt to alter the adsorption energy of CO by adjusting the *d*-band of Pt [*J. Am. Chem. Soc.*, **2006**, 128, 8813]; (2) **oxophilic effect:** This approach accelerates the removal of adsorbed CO (CO_{ad}) by supplying oxygen-containing species, also referred to as the oxophilic effect [*J. Am. Chem. Soc.*, **2005**, 127, 6819] or bifunctional effect [*ACS Catal.*, **2016**, 6, 2398] The peak potential of CO_{ad} oxidation generally depends on the surface active sites of the catalyst [*J. Phys. Chem. Lett.*, **2015**, 6, 1899].

To address the concerns raised by the reviewers, we employed in situ DRIFTS to investigate the interactions of CO with different catalysts. When commercial Pt/C was exposed to a KOH solution saturated with 100 ppm CO/H₂ (**Fig. 5f**), the DRIFTS spectra revealed bands at 1384, 1576, and 2910 cm⁻¹ corresponding to *COOH, and a band at 1900 cm⁻¹ could be indexed as bridged-adsorbed CO (B-*CO) (*ACS Catal.* **2022**, 12, 14436). In contrast, no obvious peaks of B-*CO and *COOH were observed in the DRIFTS spectra of i-ZnIn-PR/C after exposure to CO, while the bands at 1538, 1393 and 1306 cm⁻¹ corresponded to CO₃²⁻ (**Fig. 5g**), indicating that *COOH might be converted into CO₃²⁻ with *OH in water (*ACS Catal.* **2022**, 12, 14436). Furthermore, prior studies have reported that *OH reacts with *CO in the HOR reactions to eliminate CO [*Nat. Commun.* **2023**, 14, 3767; *Nano Energy* **2017**, 34, 22; *Nat. Commun.* **2022**, 13, 1596]. The classical CO electro-oxidation reaction model is the Langmuir-Hinselwood (L-H) mechanism [*Langmuir*, **2009**, 25, 13661], which can be summarized as follows:

While in alkaline media, the final product CO₂ is quickly converted into carbonate. This conclusion is consistent with our in situ infrared characterization results. For example, Zhang *et al.* reported a study using CO stripping voltammetry, and in-situ DRIFTS were used together to investigate the origin of varied CO poisoning tolerance on three Pt₃Co catalysts [*Nano Energy*, **2017**, 34, 22]. The

results illustrated that the electronic effect plays a major role in weakening CO adsorption on Pt₃Co and thus promoting CO oxidation to form COOH_{ad} intermediate consistent with the Langmuir-Hinselwood mechanism. Oxophilic effect promotes the oxidation of COOH_{ad} intermediate into the final products CO₂/CO₃²⁻. In addition, DFT calculations have been employed to demonstrate the feasibility of producing *COOH intermediates by the reaction of *CO and *OH [Angew. Chem. Int. Ed. 2021, 133, 26381]. Our DFT calculations also support this since the adsorption of *CO on i-ZnIn-PR was weaker than that on PR, suggesting that *CO on i-ZnIn-PR/C may be more readily bound to *OH. Additionally, the energy barrier for *CO oxidation to CO₂ via the intermediate *COOH on i-ZnIn-PR/C was more preferential than that on PR, as evidenced by the lower energy barrier of i-ZnIn-PR/C (0.27 eV) than that of PR (0.79 eV) (Fig. 6k). These analyses collectively confirm the successful conversion of CO to CO₂ through the formation of *COOH intermediates.

5. The reaction energies of HOR on i-ZnIn-PR and PR in Fig.6j are very strange, generally, the free energy of the final step (2H₂O+2e⁻) should be close to zero.

[Authors' Response]: Thanks very much for your kind comments. We sincerely apologize for the confusion caused by our figure demonstrations. To address this concern, we have supplied the following explanations. Currently, there is still a lack of comprehensive theoretical calculations for the HOR reaction trends. The corresponding reaction trends are calculated based on the adsorption energies of key intermediates and the mass transfer during the HOR process. In our work, for the calculation of the last step of 2H₂O+2e⁻, the formed water is adsorbed on the surface rather than a free molecule, which represents the formation of water molecules on the catalyst surface as H₂O*. To supply a better understanding of Fig. 6j, we have updated the labels of H₂O into H₂O*. The updated Fig. 6j indicates that the formation of water on the catalyst surface shows a stronger reaction trend on the i-ZnIn-PR than PtRu with a reduced energy barrier for the H₂ dissociation. After a literature review, we have noticed that the corresponding DFT calculations of HOR are very limited, especially for the reaction trend. Similar demonstrations of HOR reaction trends have been found in Nat. Commun. 2018, 9, 2235 by Park et al. In this work, the final steps are also not zero for the water formation as adsorbed 2H₂O* on the catalyst surface, where the final reaction energies are varied from -0.7 eV to -1.3 eV on different electrocatalyst surfaces. According to your comments, we have updated Fig. 6j and the discussions in the manuscript. All the revisions have been highlighted in yellow texts in the revised manuscript.

Updated Fig. 6j. The reaction energies of HOR on i-ZnIn-PR and PR.

6. The peak position of OH* in Fig 5h of in-situ Raman spectra is quite different from other studies. Besides, deuterium isotopic experiments (D₂O with KOD) should be provided for better comparison.

[Authors' Response]: Thanks very much for your kind comments. The Raman shift positions of OH adsorbed by different catalysts can vary due to a combination of factors, including catalyst composition, surface structure, chemical environment, electronic effects, defects, and adsorbate interactions. As shown in **Table R3**, we noticed that different catalysts exhibited different peak positions for the adsorption of OH*. Regarding the results reported in the literature, we can attribute 988 cm⁻¹ to OH*. As requested, the Raman deuterium isotopic experiments were given in **Supplementary Fig 46**. The updated discussions of the revised manuscript are supplied below. “We then performed some isotope experiments to identify this spectral band. In the deuteration experiment, it was found that the Raman peak of 983 cm⁻¹ was shifted to near 744 cm⁻¹ (Supplementary Fig. 46), indicating that the peak was related to the “H” atom. For species that peak at 983 cm⁻¹, such a large shift in the D₂O experiment indicates an “OH” species.²⁹”

Table R3 Comparison of *OH peak position between i-ZnIn-PR/C and other reported catalysts.

Catalyst	*OH peak position	Ref.
i-ZnIn-PR/C	983	In this work
Pt(100) surface	1090	Angew. Chem., Int. Ed. 2020 , 59, 23554
Pt(311) surface	1040	J. Am. Chem. Soc. 2020 , 142, 2, 715
Au@2.5 nmRu	715	Nat. Commun. , 2023 , 14, 5289
Au@Pt ₁ Ru _{0.2}	712~724	Nano Lett. 2022 , 22, 13, 5544
Au@PtNi	778	Angew. Chem. Int. Ed. 2021 , 60, 5708.

Supplementary Fig 46. In situ Raman spectra of i-ZnIn-PR/C collected in H₂-saturated KOD solution (0.1 M).

7. AEMFC tests with the catalysts should be provided and compared with commercial Pt/C and PtRu/C.

[Authors' Response]: Thanks very much for your kind comments. As requested, we compared

the AEMFC performance of commercial Pt/C, PtRu/C, and i-ZnIn-PR/C. The updated discussions of the revised manuscript are supplied below.

“*Impressively, i-ZnIn-PR/C exhibits excellent AEMFCs activity with a low loading amount of noble metal. Fig. 3f, h shows the comparison of AEMFCs polarization and power density curves at H₂/O₂ and H₂/Air (CO₂-free) for different anode catalysts. For H₂/O₂, the PPD of AEMFC with i-ZnIn-PR/C reaches 1.84 W cm⁻² at 4.6 A cm⁻², which is higher than the AEMFCs with commercial Pt/C (PPD of 0.85 W cm⁻² at 2.0 A cm⁻²) and PtRu/C (PPD of 1.11 W cm⁻² at 2.2 A cm⁻²). Note that the noble metal utilization of the AEMFCs is 18.4 W mg_{Pt+Ru}⁻¹, one of the highest values among the reported membrane-electrode assemblies (Fig. 3g).²⁴ Additionally, similar results was obtained in H₂/Air (CO₂-free) cell (Fig. 3h). The i-ZnIn-PR/C AEMFC delivers PPD of 1.19 W cm⁻² at 3.0 A cm⁻² under H₂-Air (CO₂-free), which is higher than the AEMFCs with commercial Pt/C (PPD of 0.67 W cm⁻² at 1.6 A cm⁻²) and PtRu/C (PPD of 0.87 W cm⁻² at 2.0 A cm⁻²).” in the revised version.*

For the MEA test, please The updated discussions of the revised manuscript are supplied below.

“*MEA test. First, the ink was prepared by mixing commercial Pt/C, PtRu/C, and i-ZnIn-PR/C with an ionomer solution composed of the homemade QAPPT dissolved in DMSO (20 wt.% QAPPT, 80 wt.% catalysts). The prepared ink was then sprayed on both sides of the alkaline polymer electrolytes (APEs, 25 ± 3 μm in thickness) to produce a catalyst-coated membrane (CCM) with the electrode area of 4 cm². And then rinsed with deionized water several times to remove the excess KOH. The resulting wet CCM is placed between two pieces of carbon paper to make the membrane electrode assembly (MEA). The H₂-O₂ fuel cells were tested using an 850E Multi Range fuel cell test station under a galvanic mode at 80 °C. The flow rate of H₂ and O₂/Air (CO₂-free) gases were all 1000 mL min⁻¹ with 0.2 MPa of backpressures. The anode and cathode noble metal loadings were 0.1 mg cm⁻² and 0.4 mg cm⁻², respectively. The prepared CCM was soaked in 1 M KOH solution at 60 °C for overnight, and rinsed with deionized water several times to remove the excess KOH. The resulting wet CCM is placed between two pieces of carbon paper to make the membrane electrode assembly (MEA). The H₂-O₂ fuel cells were tested using an 850E Multi Range fuel cell test station under a galvanic mode at 80 °C. The flow rate of H₂ and O₂/Air (CO₂-free) gases were all 1000 mL min⁻¹ with 0.2 MPa of backpressures. The anode and cathode of the humidifying temperature were set as 80 °C.” in the revised version.*

Fig. 3 HOR performance evaluation of Pt-based catalysts. **f** Fuel cell performance test of H₂/O₂ AEMFCs with commercial Pt/C (0.1 mg_{Pt} cm⁻²), PtRu/C (0.1 mg_{Pt+Ru} cm⁻²) and i-ZnIn-PR/C (0.1 mg_{Pt+Ru} cm⁻²) in anode and Pt/C (0.4 mg_{Pt} cm⁻²) in cathode. **g** Comparison of the noble metal utilization for H₂-O₂ AEMFCs. **h** Fuel cell performance test of H₂/Air (CO₂-free) AEMFC with commercial Pt/C (0.1 mg_{Pt} cm⁻²), PtRu/C (0.1 mg_{Pt+Ru} cm⁻²) and i-ZnIn-PR/C (0.1 mg_{Pt+Ru} cm⁻²) in anode and Pt/C (0.4 mg_{Pt} cm⁻²) in cathode.

In the end, we would like to thank you for the precious time of the reviewers and the editor. We sincerely hope that our point-to-point response and the revised manuscript can address your concerns and satisfy your requirements for publication. We would be grateful if we have the chance to share our work with readers of *Nature Communications*.

REVIEWER COMMENTS

Reviewer #1 (Remarks to the Author):

The authors have made great improvements to the manuscript. However, there are still some improvements that should be made before the manuscript reaches the publication criteria of Nat. Commun.

1. Could the author provide any experimental evidence supporting the distribution of oxygen as depicted in Figure 1a?
2. The authors believe that indium replaces the positions of Ru/Pt in i-ZnIn-PR. However, considering that the XAFS results for i-ZnIn-PR only show In-O bonding and the XPS results indicate solely In³⁺, how do the authors substantiate that indium in i-ZnIn-PR is doped into the RuPt rather than existing in the form of amorphous InO_x? Furthermore, even if the authors suggest that Zn and In exist in a single atom form, there also should still be a significant presence of Zn-M and In-M bonds [Nat. Commun. 10, 5812 (2019), Nat. Commun. 13, 3188 (2022)].
3. In Supplementary Figure 1c, the TEM data show nanowires with a width clearly exceeding 5 nm, which seems contradictory to other data presented; could the authors clarify this discrepancy?
4. The authors suggest that 'oxophilic metal atoms can interact with Pt and Ru to form Pt-Pt (M) and Ru-O-M coordination.' In that case, it would be beneficial for the authors to provide the XAS results of initial PtRu as a comparison to demonstrate the changes in the chemical environment within PtRu due to the introduction of the oxophilic metal atoms. It is also important to clarify whether Ru-O-M already exists in the initial PtRu. Additionally, how do the authors differentiate and confirm that in i-ZnIn-PR, the coordination is Ru-O-Zn/In rather than merely Ru-O-Ru?

Reviewer #2 (Remarks to the Author):

All my concerns have been carefully resolved by the authors. This manuscript now is ready for publication.

Reviewer #3 (Remarks to the Author):

The author addressed most of the questions I raised and it is suitable for publication in Nature Communications.

Response to Reviewers

Dear Reviewers,

Thanks very much for your precious time to constructive comments on our manuscript titled “**Interstitial metal-doped PtRu ultrathin nanowires for hydrogen oxidation catalysis**” (Manuscript ID: NCOMMS-23-33740A) for *Nature Communications*. We sincerely appreciate your comments and suggestions on our work, which are highly important for further improvements to our manuscript. According to all the comments, we have made a detailed response and substantial revisions to our revised manuscript.

Reviewer #1 (Remarks to the Author):

The authors have made great improvements to the manuscript. However, there are still some improvements that should be made before the manuscript reaches the publication criteria of Nat. Commun.

[Authors' Response]: Thanks very much for your kind support of our work. We are very grateful for your interest and appreciation of this work as well as the confirmation of its significance. Based on your valuable comments, we have made the corresponding revisions to improve our manuscript. We sincerely hope that the revised manuscript will satisfy your stringent criteria for publication.

Could the author provide any experimental evidence supporting the distribution of oxygen as depicted in Figure 1a?

[Authors' Response]: Thank you very much for this insightful comment. As requested by reviewer 1, the existence of O has been confirmed by the XANES and EXAFS spectra at Zn *K*-edge, In *K*-edge, and Ru *K*-edge, respectively. It is found that the edges of i-ZnIn-PR positively shift when compared to Zn, In, and Ru foil, suggesting that Zn, In, and Ru are slightly oxidized. Besides, the presence of In–O, Zn–O, and Ru–O in the EXAFS spectra further confirms the existence of oxides in i-ZnIn-PR. Moreover, Pt is mainly presented as the metallic state (Pt^0) based on XANES spectra of i-ZnIn-PR at Pt *L*₃-edge (Fig. 4f). Based on the above results, O atom exists with In–O–Ru and Zn–O–Ru, respectively (Supplementary Tables 8–11).

Fig. 4 Structural characterizations of i-ZnIn-PR. **a** HAADF-STEM image, **b** XRD pattern, **c-e** HRTEM images, **f** Pt L_3 -edge XANES spectra, **g** Pt L_3 -edge FT-EXAFS spectra, **h** Ru K -edge XANES spectra, **i** Ru K -edge FT-EXAFS spectra, **j** Zn K -edge XANES spectra, **k** Zn K -edge FT-EXAFS spectra, **l** In K -edge XANES spectra, **m** In K -edge FT-EXAFS spectra.

Supplementary Table 8. Structural parameters of Pt foil and i-ZnIn-PR extracted from the EXAFS fitting.

Sample	Shell	CN	R (Å)	$\sigma^2/\text{Å}^2$	R-factor
Pt foil	Pt-Pt	12	2.8 ± 0.01	0.0050 ± 0.0011	0.013
PR	Pt-Pt (Ru)	3.99 ± 2.5	2.7 ± 0.03	0.015 ± 0.003	0.018
i-ZnIn-PR	Pt-Pt (M)	6.5 ± 1.4	2.7 ± 0.02	0.0066	0.019

Note: CN is the coordination number; R is interatomic distance; σ^2 is the Debye-Waller factor (a measure of thermal and static disorder in absorber-scatterer distances); S_0^2 for Ru-Ru was set as 0.64, which is obtained from the experimental EXAFS fit of the Ru foil reference by fixing CN to the known crystallographic value and was fixed to all samples; R-factor is used to value the goodness of the fitting.

Supplementary Table 9. Structural parameters of Ru foil and i-ZnIn-PR extracted from the EXAFS fitting.

Sample	Shell	CN	R (Å)	$\sigma^2/\text{Å}^2$	R-factor
Ru foil	Ru-Ru	12	2.7 ± 0.01	0.0026 ± 0.0013	0.021
i-ZnIn-PR	Ru-O	3.8 ± 1.43	2.1 ± 0.02	0.0100 ± 0.0036	0.016
i-ZnIn-PR	Ru-Ru (M)	5.2 ± 1.4	2.7 ± 0.03	0.0181 ± 0.0040	0.016

Note: CN is the coordination number; R is interatomic distance; σ^2 is the Debye-Waller factor (a measure of thermal and static disorder in absorber-scatterer distances); S_0^2 for Ru-Ru was set as 0.64, which is obtained from the experimental EXAFS fit of the Ru foil reference by fixing CN to the known crystallographic value and was fixed to all samples; R-factor is used to value the goodness of the fitting.

Supplementary Table 10. Structural parameters of Zn foil, ZnO, and i-ZnIn-PR extracted from the EXAFS fitting.

Sample	Shell	CN	R (Å)	$\sigma^2/\text{Å}^2$	R-factor
Zn foil	Zn-Zn	6	2.6 ± 0.02	0.0130 ± 0.0011	0.019
ZnO	Zn-O	4	2.0 ± 0.03	0.0093 ± 0.0024	0.017
i-ZnIn-PR	Zn-O	3.02 ± 0.22	2.06 ± 0.03	0.0125	0.019
i-ZnIn-PR	Zn-Pt	1.61 ± 0.37	2.66 ± 0.04	0.0077	0.019

Note: CN is the coordination number; R is interatomic distance; σ^2 is the Debye-Waller factor (a measure of thermal and static disorder in absorber-scatterer distances); S_0^2 for Zn-Zn was set as 0.70, which is obtained from the experimental EXAFS fit of the Zn foil reference by fixing CN to the known crystallographic value and was fixed to all samples; R-factor is used to value the goodness of the fitting.

Supplementary Table 11. Structural parameters of In₂O₃ and i-ZnIn-PR extracted from the EXAFS fitting.

Sample	Shell	CN	R (Å)	$\sigma^2/\text{Å}^2$	R-factor
In ₂ O ₃	In–O	6	2.17 ± 0.01	0.0059 ± 0.0016	0.004
i-ZnIn-PR	In–O	5.21 ± 1.26	2.16 ± 0.03	0.0060 ± 0.0036	0.019
i-ZnIn-PR	In–Pt	0.54 ± 0.04	3.23 ± 0.13	0.0021 ± 0.0037	0.019

Note: CN is the coordination number; R is interatomic distance; σ^2 is the Debye-Waller factor (a measure of thermal and static disorder in absorber-scatterer distances); S_0^2 for In–O was set as 0.99, which is obtained from the experimental EXAFS fit of the In₂O₃ reference by fixing CN to the known crystallographic value and was fixed to all samples; R-factor is used to value the goodness of the fitting.

The authors believe that indium replaces the positions of Ru/Pt in i-ZnIn-PR. However, considering that the XAFS results for i-ZnIn-PR only show In–O bonding and the XPS results indicate solely In³⁺, how do the authors substantiate that indium in i-ZnIn-PR is doped into the RuPt rather than existing in the form of amorphous InO_x? Furthermore, even if the authors suggest that Zn and In exist in a single atom form, there also should still be a significant presence of Zn–M and In–M bonds [*Nat. Commun.* 10, 5812 (2019), *Nat. Commun.* 13, 3188 (2022)].

[Authors' Response]: Thank you very much for this valuable comment. We concluded that In was doped into RuPt based on the analysis of XANES and EXAFS spectra. As commented by the Reviewer, if In was presented as InO_x, the features of In–O (the first shell) and In–In (the second shell) coordination would be present in the EXFAS spectrum of i-ZnIn-PR at In K-edge. As shown in Fig. 4m, the strong features of In–O coordination and In–In coordination were observed in the EXAFS spectrum of In₂O₃ reference. For i-ZnIn-PR, the feature of In–In coordination almost disappeared in the EXAFS spectrum, while the weaker feature with a shorter distance was ascribed to Pt–In (**Fig. 4m**) due to the different electronegativity of Pt and In.

We completely agree with you that the features of Zn–M and In–M should be present in the EXAFS spectrum, and we labeled the features Pt–Zn and Pt–In in **Fig. 4k** and **Fig. 4m**, respectively. In addition, we analyzed the Pt–Zn and Pt–In by fitting the EXAFS spectra at Zn K-edge and In K-edge (**Supplementary Tables 10 and 11**).

Fig. 4 Structural characterizations of i-ZnIn-PR. **a** HAADF-STEM image, **b** XRD pattern, **c-e** HRTEM images, **f** Pt L_{3} -edge XANES spectra, **g** Pt L_{3} -edge FT-EXAFS spectra, **h** Ru K -edge XANES spectra, **i** Ru K -edge FT-EXAFS spectra, **j** Zn K -edge XANES spectra, **k** Zn K -edge FT-EXAFS spectra, **l** In K -edge XANES spectra, **m** In K -edge FT-EXAFS spectra.

Supplementary Table 10. Structural parameters of Zn foil, ZnO, and i-ZnIn-PR extracted from the EXAFS fitting.

Sample	Shell	CN	R (\AA)	$\sigma^2/\text{\AA}^2$	R-factor
Zn foil	Zn-Zn	6	2.6 ± 0.02	0.0130 ± 0.0011	0.019
ZnO	Zn-O	4	2.0 ± 0.03	0.0093 ± 0.0024	0.017
i-ZnIn-PR	Zn-O	3.02 ± 0.22	2.06 ± 0.03	0.0125	0.019
i-ZnIn-PR	Zn-Pt	1.61 ± 0.37	2.66 ± 0.04	0.0077	0.019

Note: CN is the coordination number; R is interatomic distance; σ^2 is Debye-Waller factor (a measure of thermal and static disorder in absorber-scatterer distances); S_0^2 for Zn-Zn was set as 0.70, which is obtained from the experimental EXAFS fit of the Zn foil reference by fixing CN to the known crystallographic value and was fixed to all samples; R-factor is used to value the goodness of the fitting.

Supplementary Table 11. Structural parameters of In₂O₃ and i-ZnIn-PR extracted from the EXAFS fitting.

Sample	Shell	CN	R (\AA)	$\sigma^2/\text{\AA}^2$	R-factor
In ₂ O ₃	In-O	6	2.17 ± 0.01	0.0059 ± 0.0016	0.004
i-ZnIn-PR	In-O	5.21 ± 1.26	2.16 ± 0.03	0.0060 ± 0.0036	0.019
i-ZnIn-PR	In-Pt	0.54 ± 0.04	3.23 ± 0.13	0.0021 ± 0.0037	0.019

Note: CN is the coordination number; R is interatomic distance; σ^2 is the Debye-Waller factor (a measure of thermal and static disorder in absorber-scatterer distances); S_0^2 for In-O was set as 0.99, which is obtained from the experimental EXAFS fit of the In₂O₃ reference by fixing CN to the known crystallographic value and was fixed to all samples; R-factor is used to value the goodness of the fitting.

In Supplementary Figure 1c, the TEM data show nanowires with a width clearly exceeding 5 nm, which seems contradictory to other data presented; could the authors clarify this discrepancy?

[Authors' Response]: Thank you very much for carefully reviewing our manuscript. We apologize for this mistake in the scale bar and have updated **Supplementary Figure 1**.

Supplementary Figure 1. (a) TEM image, (b, c) HRTEM images, and (d) SEM-EDS profile of PR. Inset of (a) presents the diameter distribution of PR.

The authors suggest that 'oxophilic metal atoms can interact with Pt and Ru to form Pt–Pt (M) and Ru–O–M coordination.' In that case, it would be beneficial for the authors to provide the XAS results of initial PtRu as a comparison to demonstrate the changes in the chemical environment within PtRu due to the introduction of the oxophilic metal atoms. It is also important to clarify whether Ru–O–M already exists in the initial PtRu. Additionally, how do the authors differentiate and confirm that in *i*-ZnIn-PR, the coordination is Ru–O–Zn/In rather than merely Ru–O–Ru? **[Authors' Response]:** Thank you for this insightful comment. As requested, we have provided the XAS results of the initial PtRu. The absence of Ru–O–M feature in the EXAFS spectrum of PtRu suggests that Ru–O–M is attributed to the introduction of oxophilic metal atoms.

Regarding the comment “*how do the authors differentiate and confirm that in i-ZnIn-PR, the coordination is Ru–O–Zn/In rather than merely Ru–O–Ru?*”, it is found that the distance of Ru–O in *i*-ZnIn-PR is slightly longer than that in RuO₂ (**Fig. 1h**), which means that the second shell is not only Ru atom in the form of Ru–O–Ru. Given the stronger electronegativity of In and Zn than Ru, we conclude that the long distance of Ru–O in *i*-ZnIn-PR is attributed to the 2nd shell of the In/Zn atom, which stretches the Ru–O distance.

Please kindly check the revised sentence as follows.

“As shown in Fig. 1e, the Pt L₃-edge XANES spectrum of PR and *i*-In-PR displayed similar features to that of Pt foil, indicating that Pt mainly presented as metallic state in PR and *i*-In-PR, which was further validated by the presence of Pt–Pt (Ru/In) coordination in EXAFS spectrum (Fig. 1f) and wavelet transform pattern (Supplementary Fig. 4a).¹⁸ Analysis on Ru K-edge XANES spectrum suggested that Ru in *i*-In-PR presented as oxidative state (Ru^{δ+}, 0 < δ < 4) (Fig. 1g),¹⁹ which was

further confirmed by the presence of Ru–O in EXAFS spectrum (Fig. 1h and Supplementary Fig. 4b). Note that the radius of Ru–O in *i*-In-PR was larger than that of RuO₂ and PR, which was attributed to the formation of Ru–O–M (M = In or Zn) with stronger electronegativity than Ru.²⁰ in the revised manuscript.

Fig. 1 Characterizations of *i*-M-PR. **a** Scheme for introducing foreign atoms into PR. **b** HAADF-STEM image, **c** XRD pattern. **d** Aberration-corrected HAADF-STEM images, **e** Pt *L*₃-edge XANES spectra, **f** Pt *L*₃-edge FT-EXAFS spectra, **g** Ru *K*-edge XANES spectra, **h** Ru *K*-edge FT-EXAFS spectra of *i*-In-PR. **i** Pt XANES and **j** FT-EXAFS spectra of *i*-M-PR at *L*₃-edge.

Reviewer #2 (Remarks to the Author):

All my concerns have been carefully resolved by the authors. This manuscript now is ready for publication.

[Authors' Response]: Thanks very much for your kind support and recognition of our work. We highly appreciate your precious time and great efforts in reviewing our work. Your interest and detailed comments on our work are a great encouragement for our research as well as future works.

Reviewer #3 (Remarks to the Author):

The author addressed most of the questions I raised and it is suitable for publication in Nature Communications.

[Authors' Response]: Thanks very much for your kind support of our work. We highly appreciate your time and efforts in reviewing our work. All your comments are very constructive for improving our research.

In the end, we would like to thank you for the precious time of the reviewers and the editor. We sincerely hope that our point-to-point response and the revised manuscript can address your concerns and satisfy your requirements for publication. We would be grateful if we have the chance to share our work with readers of *Nature Communications*.

REVIEWERS' COMMENTS

Reviewer #1 (Remarks to the Author):

The authors have addressed all the comments. No more revisions are required.